# Determination of Local Stresses and Strains within the Notch Strain Approach: The Efficient and Accurate Calculation of Notch Root Strains Using Finite Element Analysis

**Lukas Masendorf *** , **Ralf Burghardt** , **Michael Wächter** and **Alfons Esderts**

Institute for Plant Engineering and Fatigue Analyses, Clausthal University of Technology, Leibnizstraße 32, 38678 Clausthal-Zellerfeld, Germany; ralf.burghardt@tu-clausthal.de (R.B.); michael.waechter@tu-clausthal.de (M.W.); alfons.esderts@tu-clausthal.de (A.E.)
* Correspondence: lukas.masendorf@tu-clausthal.de

**Abstract:** For the service life estimation of metallic components under cyclic loading according to strain-based approaches, a simulation of the elastic-plastic stress–strain path at the point of interest is necessary. An efficient method for determining this stress–strain path is the use of the load–notch-strain curve, as this is also implemented within the FKM guideline nonlinear. The load–notch-strain curve describes the relationship between the load on the component and the local elastic-plastic strain. On the one hand, this can be estimated from loads or theoretical elastic stresses by using notch root approximations. On the other hand, this can be determined in a finite element analysis based on the elastic-plastic material behaviour. This contribution describes how this latter option is carried out in general and how it can be optimised in such a way that the FEA requires significantly less calculation time. To show the benefit of this optimisation, a comparative calculation on an exemplary geometry is carried out.

**Keywords:** service life estimation; load–notch-strain curve; notch strain approach; finite element analysis

## 1. Introduction

The assessment of fatigue strength is one of the most important aspects in the design of safety-related components. As an alternative or as a supplement to experimental strength assessments, analytical assessments can be performed. Various calculation approaches exist for this purpose; these can be distinguished into stress- and strain-based concepts [1,2]. The latter, which are also called local strain approaches or the notch strain concept, assume elastic-plastic material behaviour in the component, and use the local stresses and strains calculated as basic load quantities.

The notch strain concept, also known as the local strain approach, can be found in many variants throughout the literature [1–5]. These differ primarily in the procedures used to determine the local stresses and strains; e.g., the notch root approximations [6–16], and in the so-called load parameters (also known as mean stress parameters or damage parameters) used to evaluate the damage of individual stress–strain hystereses [17–24]. Both can have a significant influence on the calculation result of a component's fatigue life. In addition, the calculation procedures used for the notch strain approach—regardless of how it is expressed—can only be applied with the help of computational algorithms, since, for example, numerical solution procedures need to be used. Therefore, the calculation result also depends to a certain extent on the user-specific implementation of these calculation algorithms.

In order to use local strain approaches for a reliable analytical component assessment, the above-mentioned diversity of variants and dependency of the calculation result on the implementation tend to be disadvantages. Calculation results are not comparable. To overcome this weakness, the "Rechnerischer Festigkeitsnachweis für Maschinenbauteile

unter expliziter Erfassung nichtlinearen Werkstoffverformungsverhaltens" (Analytical strength assessment for components under explicit consideration of nonlinear material behaviour [25]) guideline was developed in Germany. In this guideline, the above-mentioned diversity of variants in the calculation algorithms is reduced to two variants with different levels of complexity. Both variants are also described in such detail that the deviations in the calculation results of different users are reduced to a minimum. In addition, the calculation algorithm is provided with a safety factor concept that allows the assessment of component fatigue lives for low probabilities of failure. The described guideline was developed by the German research association "Forschungskuratorium Maschinenbau" (FKM), and subsequently will be referred to as the "FKM guideline nonlinear". As the guideline is only available in the German language at present, the calculation model for the fatigue strength assessment will be briefly explained below. More detailed information can be found in [25,26].

The FKM guideline nonlinear allows two different calculation procedures for fatigue strength assessment that differ in their estimation of the local stresses and strains and the load parameter used.

- On the one hand, the local elastic-plastic stresses and strains can be estimated using the notch root approximation according to Neuber [27] with the extension according to Seeger and Heuler [28]. For each closed hysteresis detected from the local stress–strain paths estimated in this way, a load parameter $P_{RAM}$ is calculated using Equation (1):

$$
\begin{aligned}
P_{RAM} &= \sqrt{(\sigma_a + k \cdot \sigma_m) \cdot \varepsilon_a \cdot E} && \text{if } (\sigma_a + k \cdot \sigma_m) \geq 0 \\
P_{RAM} &= 0 && \text{if } (\sigma_a + k \cdot \sigma_m) < 0
\end{aligned}
\tag{1}
$$

where $\sigma_a$ is the stress amplitude, $\sigma_m$ is the mean stress, $\varepsilon_a$ is the strain amplitude of the detected hysteresis, $E$ is Young's modulus, and $k$ is a correction factor that takes into account the mean stress sensitivity of the material. The load parameter $P_{RAM}$ is a modification of the widely used approach according to Smith, Watson, and Topper [17]. This modification extends the Smith, Watson, and Topper approach using the material-dependent mean stress sensitivity, as originally suggested by Bergmann [19,29]. With the help of the $P_{RAM}$ values for the individual hysteresis and a corresponding $P_{RAM}$ Wöhler curve, damage accumulation can be carried out, and the component service life can be calculated.
- On the other hand, the local stresses and strains can be estimated using the notch root approximation according to Seeger and Beste [30–32]. The evaluation of the damage of the individual hysteresis is carried out with the $P_{RAJ}$ load parameter, which is a refined version of the $P_J$ parameter according to Vormwald [21,24]:

$$
P_{RAJ} = 1.24 \cdot \frac{\Delta\sigma_{eff}^2}{E} + \frac{1.02}{\sqrt{n'}} \cdot \Delta\sigma_{eff} \cdot \Delta\varepsilon_{eff} - \frac{\Delta\sigma_{eff}}{E}
\tag{2}
$$

This is motivated by the fracture mechanics of short cracks, and is thus able to take into account the sequence effects of different load sequences through crack opening and closing effects.

The calculation procedure based on $P_{RAM}$ is less complex in terms of the calculation algorithm, and therefore easier to implement. However, the variant using $P_{RAJ}$ has the advantage that load sequence effects can be taken into account, which promises a better accuracy, especially for load sequences with changing statistic characteristics.

Both described calculation procedures within the FKM guideline nonlinear have in common the fact that notch root approximations are used as standard approaches for determining the local stresses and strains. As an alternative approach, the use of finite element analyses (FEA) with elastic-plastic material behaviour is also permitted. However, detailed instructions on how to proceed with such an analysis in order to calculate the local stresses and strains both efficiently and accurately are lacking. While another contribution

by the authors deals with the efficient implementation of notch root approximations to determine local stresses and strains [33], this article focuses on the aspect of using FEA.

The FEA with elastic-plastic material behaviour has the advantage that all influences of the geometry (stiffness distribution, cross-section changes) can be considered in the analysis. Compared to the notch root approximation, this procedure promises a higher accuracy in the determination of the local stresses and strains independent of the considered geometry, and thus a higher accuracy in the service lives calculated with the notch strain concept. A disadvantage of this is the significantly higher calculation effort required in terms of computational resources and calculation time compared to FEA with linear-elastic material behaviour and a subsequent notch root approximation, especially when finite element models with many elements are considered—e.g., for complex geometries.

This is also the reason why the standard use of FEA with elastic-plastic material behaviour to determine local stresses and strains has not yet found its way into regulations for the assessment of fatigue strength. Rather, this sort of finite element analysis is only used in individual cases [34–40], in which the calculation time only plays a subordinate role, or for the development of notch root approximations themselves [7,14,15,41,42], where the stresses and strains determined in this way are used as a reference for those estimated with notch root approximations. Recommendations for the efficient design of finite element analyses with which input data for strain-based strength assessments can be generated can only be found in rudimentary form—see, e.g., [43].

This paper therefore presents a procedure for determining load–notch-strain curves (Section 2) using FEA. The focus is placed in particular on the following aspects of how to increase the efficiency of the calculation:

- The discretisation of the used material law;
- The way in which nodal results are calculated;
- A reduction in the number of applied load steps.

The mentioned aspects have an effect on the determined local stresses as well as the local strains, which makes it difficult to assess their relevance to the calculated service life. For this reason, the influence of the aforementioned aspects is rated based on the calculated service lives for both constant and variable amplitude loading. The above-mentioned FKM guideline nonlinear is used for the calculation of these component service lives.

Even though the well-documented calculation algorithm of the FKM guideline nonlinear is used for the efficiency rating, the procedure recommended in this paper for determining the load–notch-strain curve can easily be applied to other variants of strain-based concepts, and is not explicitly linked to the use of the FKM guideline nonlinear.

## 2. Determination of Local Stresses and Strains for the Notch Strain Approach Using FEA

In this section, the necessary basics for understanding this contribution are described. Section 2.1 explains the elastic-plastic deformation behaviour of metallic materials that forms the basis for the following investigations. Subsequently, Section 2.2 introduces the load–notch-strain curve and its application in the strain-based concept, and Section 2.3 describes the principal determination of the load–notch-strain curve with FEA.

### 2.1. Cyclically Stabilised Uniaxial Material Behaviour

For service life estimations carried out by means of the notch strain approach, the local stress–strain path is determined under consideration of elastic-plastic material behaviour [1,2]. The uniaxial relationship between stress $\sigma$ and strain $\varepsilon$ under cyclic loading is described by a material law. For the simplification of a cyclically stabilised deformation behaviour, the cyclic stress–strain curve can be described with the approach according to Ramberg and Osgood [44], independent from the prehistory of cyclic loading (see Equation (3)).

$$\varepsilon = \frac{\sigma}{E} + \left(\frac{\sigma}{K'}\right)^{\frac{1}{n'}} \tag{3}$$

The approach of Ramberg and Osgood is also used in the FKM guideline nonlinear [25]. Young's modulus E is a fixed value in [25] depending on the material group involved—for example, E = 206 GPa for steel. The cyclic material parameters K′ and n′ can either be determined experimentally or estimated. In the FKM guideline nonlinear, the estimation method according to Wächter et al. [45,46] is given for estimating the parameters of the cyclic stress–strain curve from the tensile strength $R_m$. For steel materials, this estimation is summarised in Equations (4) and (5):

$$n' = 0.187 \tag{4}$$

$$K' = \frac{3.1148 \text{ MPa} \cdot \left(\frac{R_m}{\text{MPa}}\right)^{0.897}}{\left(\min\left[0.338; \ 1033 \cdot \left(\frac{R_m}{\text{MPa}}\right)^{-1.235}\right]\right)^{n'}} \tag{5}$$

Other approaches for describing the uniaxial cyclically stabilised material behaviour might also be utilized; e.g., a bilinear [2] or multilinear approach, or the modified Ramberg–Osgood approach according to Pei and Dong [47]. However, these approaches will not be considered in more detail in the following investigations, since the focus is on the FKM guideline nonlinear.

The simplified assumption of cyclically stabilised material behaviour is often made within analytical fatigue assessments using strain-based concepts; e.g., [2,48–51]. This is mainly motivated by the fact that the number of material parameters required for the material model used can be kept within a reasonable range for a practical application. Throughout the literature, however, there are various complex approaches that promise to represent the physical cyclic material behaviour more accurately—see, e.g., [51–53]. Assuming a cyclically stabilised material behaviour, the deformation of the material during initial loading is described by Equation (3). The stress–strain path at load reversal is described according to Masing's behaviour [54], with the initial load curve doubled in stress and strain—see Equation (6) and Figure 1A. The doubled initial load curve is referred to as a hysteresis branch below.

$$\Delta\varepsilon = \frac{\Delta\sigma}{E} + 2 \cdot \left(\frac{\Delta\sigma}{2 \cdot K'}\right)^{\frac{1}{n'}} \tag{6}$$

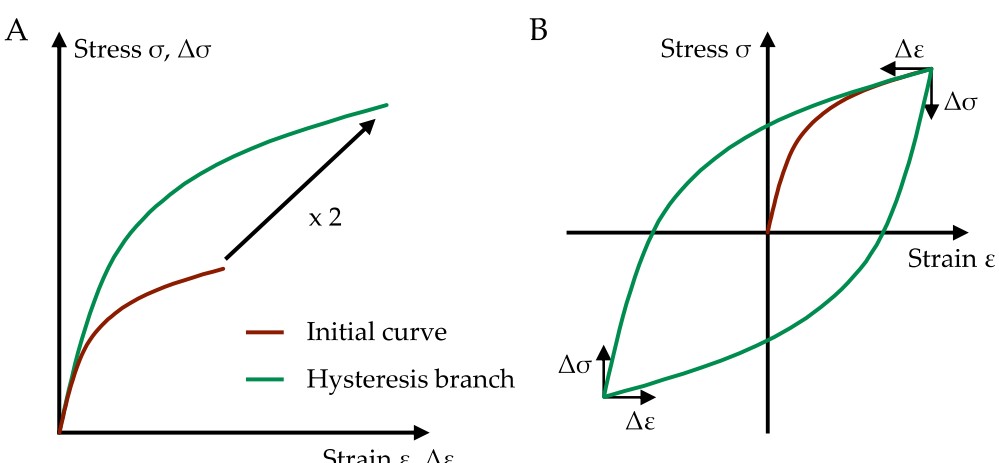

**Figure 1.** Cyclically stabilised material behaviour: (**A**) Masing's behaviour; (**B**) local stress–strain path at constant amplitude loading.

In the local stress–strain diagram, a new coordinate system for the range of stress Δσ and strain Δε is placed in each reversal point of the stress–strain curve and a new hysteresis

branch starts from there, which follows Equation (6). Its length depends on the level of stress. Figure 1B schematically shows the stress–strain path for constant amplitude loading.

If the loading on a material shows variable amplitudes, the rules of material memory (memory rules) must also be taken into account; these describe at which points of the course the stress–strain path changes from an initial load curve to a hysteresis branch, and vice versa. The memory rules have a considerable influence on the stress–strain path, and thus on the size of the closed hysteresis and the position of the hysteresis in the stress–strain diagram. The model used for the description of memory effects and the simulation of qualitatively correct local stress–strain paths was developed by various contributions—see, e.g., [55–61]. The separation into three easily understandable memory rules, which are also used in the FKM guideline nonlinear [25], is described in the work of Bergmann, Clormann, and Seeger [29,62]; see Figure 2.

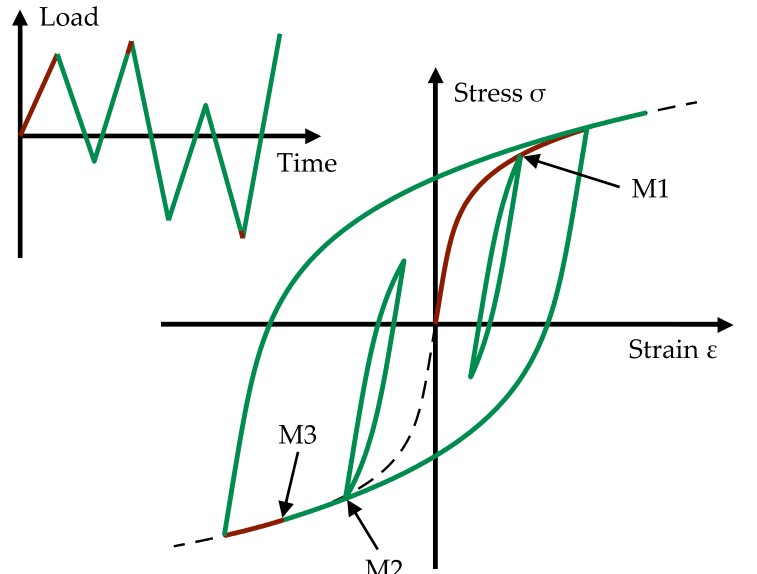

M1: After closing a hysteresis started on the initial load curve, the stress-strain path continues on the initial load curve

M2: After closing a hysteresis started on a hysteresis branch, the stress-strain path continues on the original hysteresis branch.

M3: A hysteresis branch started on the initial load curve ends when the mirror point of its starting point is reached in the opposite quadrant. The stress-strain path continues on the initial load curve.

**Figure 2.** Memory rules.

### 2.2. Simulation of Local Stress–Strain Paths and Load–Notch-Strain Curve

To estimate the service life of a component according to the notch strain approach, it is necessary to convert the sequence of the external load (e.g., forces or moments) into a path of local elastic-plastic stresses and strains at the point of interest of the component. The dependency of local stresses and strains follows the principles given in Section 2.1 as long as the material is loaded in a uniaxial stress state. For multiaxial stress states, more or less complex hypothesis and material models need to be used. The FKM guideline nonlinear assumes that multiaxial proportional loading may also be described by the same principles as those for uniaxial loading (Section 2.1) if equivalent stresses according to the von Mises hypotheses are used [63,64]. Since equivalent stresses are always positive, a sign needs to be added in this case [65]. The FKM guideline nonlinear suggests using the sign of hydrostatic stress $\sigma_H$:

$$\sigma_V = \frac{\text{sign}(\sigma_H)}{\sqrt{2}} \sqrt{(\sigma_{xx} - \sigma_{yy})^2 + (\sigma_{xx} - \sigma_{zz})^2 + (\sigma_{zz} - \sigma_{yy})^2 + 6(\tau_{xy}^2 + \tau_{yz}^2 + \tau_{zx}^2)} \quad (7)$$

$$\sigma_H = \frac{1}{3} \cdot (\sigma_{xx} + \sigma_{yy} + \sigma_{zz}) \quad (8)$$

For now, only components may be evaluated using the FKM guideline nonlinear [25] with uniaxial and multiaxial proportional loadings. This is due to the fact that neither the procedure described below of using a load–notch-strain curve to determine the stress–strain

path cannot be applied to nonproportional loads, nor is a satisfactory consideration of the effects occurring under nonproportional loads, such as nonproportional hardening [66], or life extension or shortening, cf. e.g., [67], implemented within the FKM guideline nonlinear yet.

For the application of a strain-based approach, besides the material behaviour, the dependency between the load (e.g., force, moment) and the local stress and strain needs to be known so that the local stress–strain path can be determined. Different procedures are available for determining the local stress–strain path:

- Reversal point to reversal point with FEA: The most obvious option is to transfer the component into a finite element model, to deposit an elastic-plastic material behaviour according to Section 2.1, and apply the load-time history to the component in order to obtain the local stress–strain path. Since FEAs with elastic-plastic material behaviour— and especially with complex component geometries—still require large computing capacities and thus lead to high computing times, this procedure is only recommended for very short load-time histories. Section 5 illustrates this procedure with an example.

- Reversal point to reversal point with notch root approximation: Significant time can be saved by using notch root approximations. These make it possible to estimate stresses and strains with elastic-plastic material behaviour from external loads or local elasticity-theoretical stress. The principle procedure is as follows [68]: the elastic-plastic stress–strain path is obtained by applying a notch root approximation to each reversal point of the load sequence or the elasticity-theoretical stress history, considering the memory rules [29,62] and Masing's behaviour [54].

- Use of a load–notch-strain curve: With the previously described methods, a notch root approximation or an FEA must be carried out for each reversal point of the load-time history. This leads to a high calculation effort for long load-time histories, which can be shortened by using the so-called load–notch-strain curve. The load–notch-strain curve describes the relationship between the external load or local elasticity-theoretical stress and the local elastic-plastic strain. If the previously mentioned assumption of cyclically stabilised material behaviour is made, the relationship between an external load and the local elastic-plastic stresses and strains is always the same for a single load branch. The load–notch-strain curve can be regarded as a type of template. The load–notch-strain curve may therefore be determined before the simulation of the local stress–strain path itself using either a notch root approximation or FEA. For longer load-time histories with a large number of reversal points, this procedure is much less time-consuming than a point-by-point FEA or a point-by-point notch root approximation. The idea of using a load–notch-strain curve instead of a point-by-point calculation can most likely be traced to Williams et al. [48], and was advanced by the work of [29]. In the following, the procedure for determining and using the load– notch-strain curve is described using the procedure of the FKM guideline nonlinear.

For the application of the FKM guideline nonlinear, the load-time history is classified, which means that only a discretised load–notch-strain curve is required. The class width for the discretisation is determined by dividing the largest load $L_{max}$ by the number of classes. The FKM guideline nonlinear suggests using a total of 100 classes between $L = 0$ and $L = L_{max}$; whereas in Figure 3A, a class number of 4 is used for didactic reasons. In the classification, the load at each reversal point is shifted to the next largest class limit in terms of absolute value; see the red points in Figure 3A.

In order to simplify the representation and make it independent of different units of load, as well as to correspond to the format of the FKM guideline nonlinear, the load–notch-strain curve is represented in this paper as the relationship between the linear-elastic stress and the notch strain. The same is true for the load-time history, which is also understood to be local linear-elastic stress. However, the load–notch-strain curve can also be used as a relation between external load and notch-strain.

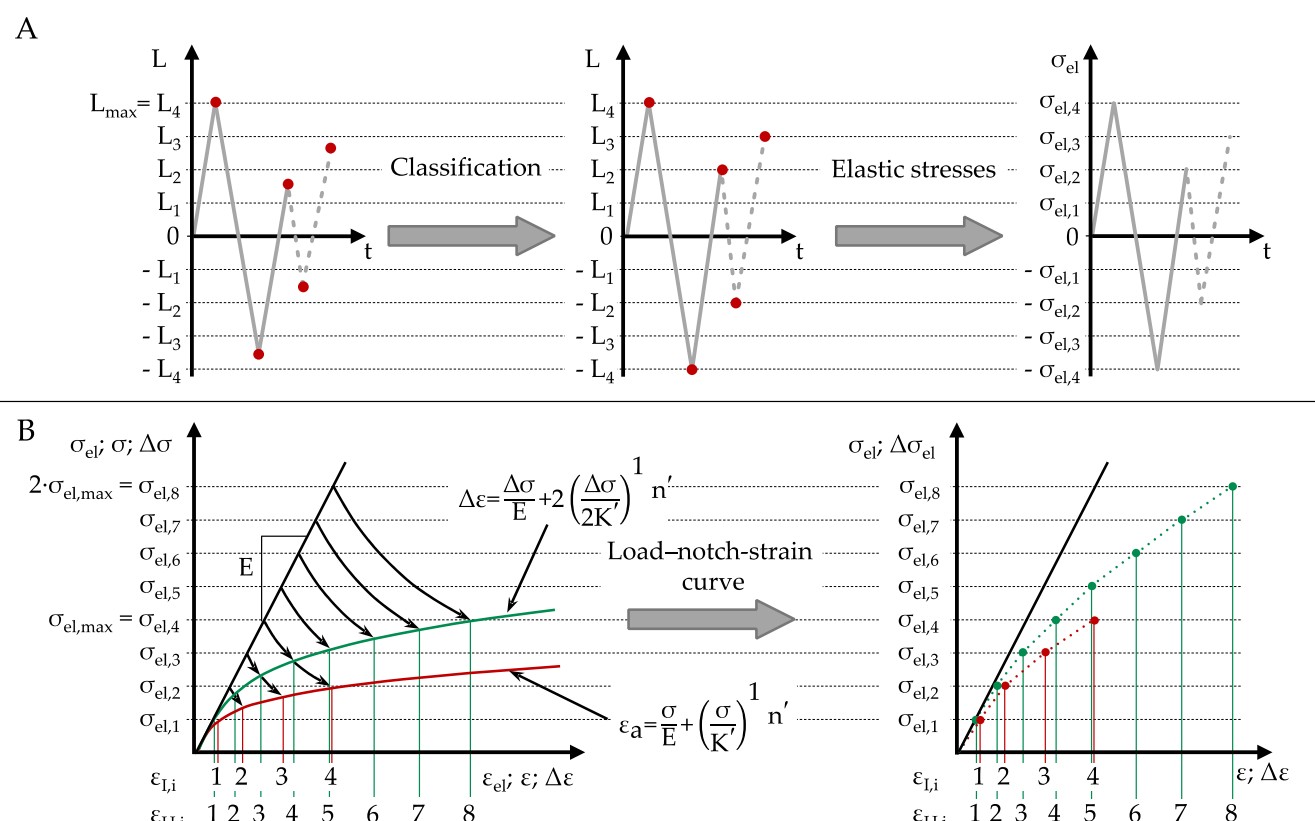

**Figure 3.** Load–notch-strain curve: (**A**) classification of the load-time history and transfer in elasticity-theoretical stresses; (**B**) determination of the load–notch-strain curve for the initial load and the hysteresis branch.

Subsequently, the load–notch-strain curve for the initial load is determined as discrete points for each class of linear-elastic stresses—i.e., the elastic-plastic stress and strain must be calculated for each class of linear-elastic stress. Figure 3B shows the qualitative calculation of the elastic-plastic strains and stresses with a notch root approximation. From the four elasticity theoretical stresses $\sigma_{el,1...4}$ at the class limits, which lie on a straight line with the slope of the elastic modulus E in the stress–strain diagram (Hooke's law), the elastic-plastic stresses, which must lie on the cyclic stress–strain curve of the material (shown in red), are inferred. The load–notch-strain curve is obtained by plotting the elasticity-theoretical stress amplitudes at the class limits $\sigma_{el,1...4}$ over the corresponding strains $\varepsilon_{I,1...4}$ for the initial load.

Due to Masing's behaviour, the local stress–strain path after load reversal does not follow the shape of the initial load curve (Equation (3)); instead, it follows the double initial load curve, the so-called hysteresis branch (Equation (6)). Therefore, in addition to the load–notch-strain curve for the initial load, the load–notch-strain curve for the hysteresis branch must also be determined. Within the load sequence, maximum stress ranges $\Delta\sigma_{el}$, which correspond to twice the maximum elasticity-theoretical stress $\sigma_{el,max}$, can occur. The load–notch-strain curve for the hysteresis branch is therefore determined for elasticity-theoretical stresses of up to $2\cdot\sigma_{el,max}$, and is shown in green in Figure 3B.

The local elastic-plastic stress–strain path can now be determined without much computational effort. In Figure 4A, the path is determined up to the first reversal point. Since this is the first load on the material, the load–notch-strain curve for the initial load is used. The amplitude of the first flank corresponds to the stress of the fourth class $\sigma_{el} = \sigma_{el,4}$. The corresponding elastic-plastic strain $\varepsilon_1$ is determined with the load–notch-strain curve and corresponds to the strain $\varepsilon_{I,4}$. The local stress–strain curve begins at the origin and continues to the strain $\varepsilon = \varepsilon_1$. The corresponding stress $\sigma$ and the course of the $\sigma$-$\varepsilon$ path are determined with the material law—see Equation (3).

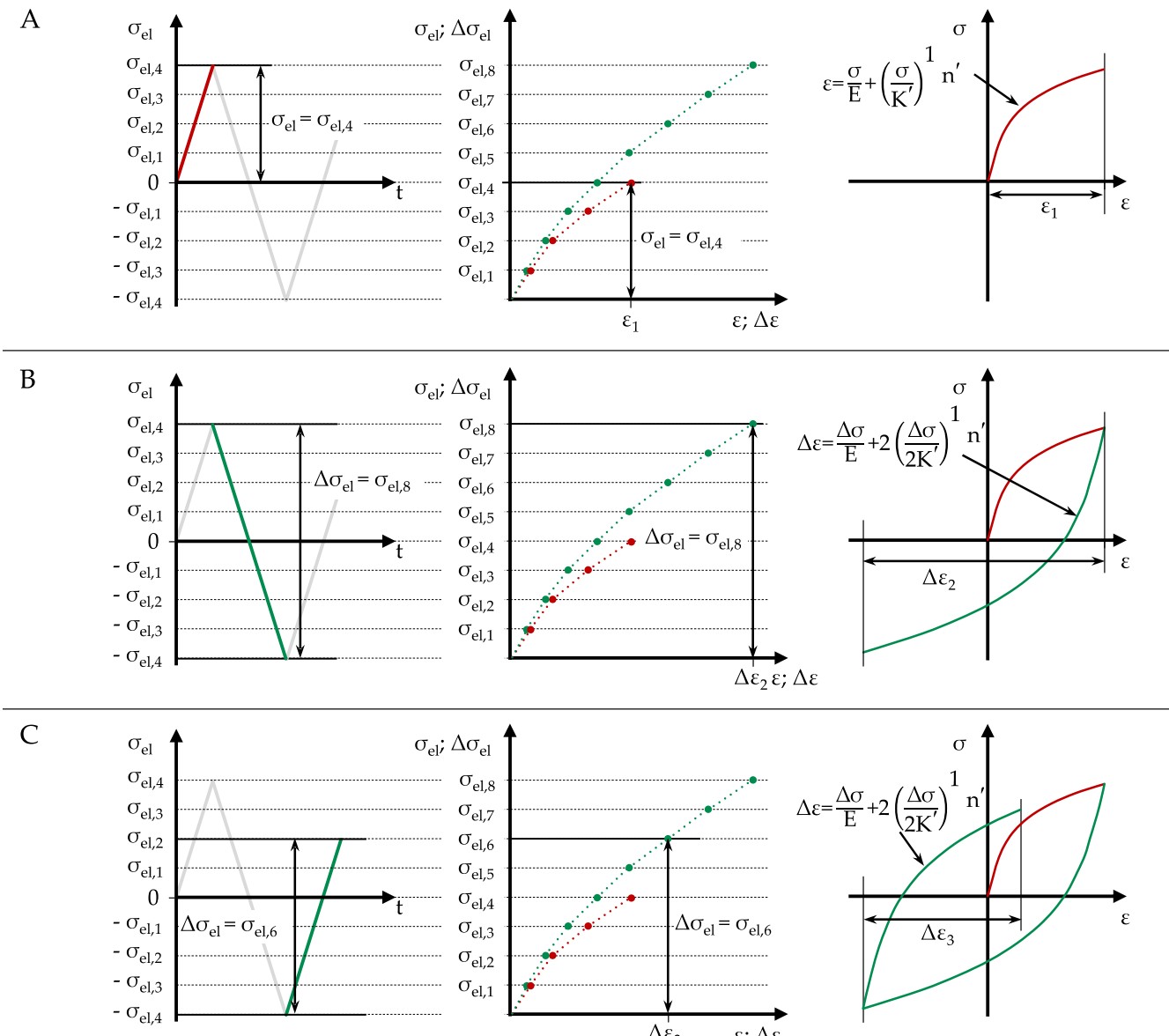

**Figure 4.** Generating the local stress–strain path using a load–notch-strain curve.

Figure 4B shows the calculation up to the next reversal point. Due to Masing's behaviour at load reversal, the load–notch strain curve for the hysteresis branch is used. The stress range of the flank corresponds to the eighth class's stress $\Delta\sigma_{el} = \sigma_{el,8}$. The associated elastic-plastic strain $\Delta\varepsilon_2$ is determined with the load–notch-strain curve. The local stress–strain curve is continued with a hysteresis branch (Equation (6)) with a stress range of $\Delta\varepsilon_2$. The determination of the local curve for the third flank is conducted in Figure 4C analogous to Figure 4B.

The simulation of the local stress–strain path is carried out in connection with the Rainflow–HCM algorithm [62], whereby the material memory is also taken into account when using a load–notch-strain curve, cf. [29].

Subsequently, the local stress–strain path is examined for closed hystereses, and these are evaluated with a load parameter in order to estimate the service life of the component. Since the focus of this contribution is the determination of the load–notch-strain curve, the service life estimation is not discussed in detail.

The FKM guideline nonlinear uses the notch root approximation according to Neuber [27] extended by Seeger and Heuler [28] or the one according to Seeger and Beste [30–32]

in order to convert linear-elastic stresses to elastic-plastic strains (for the calculation of load–notch-strain curve). Both notch root approximations depend on the knowledge of the limit load factor $K_p$ to account for the global deformation behaviour occurring due to net section plasticity [28]. The limit load factor describes how much the elastic limit load can be increased before plastic collapse occurs when the plastic limit load is reached. The yield initiation load is the load that leads to the yield point being reached in the case of elastic-ideal-plastic material behaviour. The plastic limit load is the load that occurs when the load is increased above the elastic limit load until the entire component cross-section is plastified and plastic collapse occurs. $K_p$ may be determined either analytically for simple geometries or by using FEA with elastic-ideal-plastic material behaviour for complex geometries [28]. While using FEA for the determination, the load for the plastic collapse of the component in question needs to be approximated. This means that the load is increased until no more equilibrium is found for the finite element problem. The calculation times up to this point can become quite large. This is especially noteworthy since the quantity $K_p$ is only used to correct the approximation methods of the previously mentioned notch approximations.

As an alternative to the notch approximations in connection with the $K_p$ variable, which is difficult to determine, FEA with elastic-plastic material behaviour can also be used directly, this time using the material behaviour according to Equation (3). This procedure is described below.

*2.3. Determination of Load–Notch-Strain Curves Using Finite Element Analyses*

In case the load–notch-strain curve is determined using FEA, the material behaviour of the simulation is defined according to Ramberg and Osgood (Equation (3)).

As described in Section 2.2, the load–notch-strain curve represents the relationship between the external loads or elasticity-theoretical stresses and the elastic-plastic strains at each class limit. The most obvious way to determine these elastic-plastic strains is to apply the loads at the class limits to the FE model step by step.

FEA is now conducted for each class of linear elastic stresses and the corresponding loads are applied at the class limits. If the loads or theoretical elastic stresses are plotted against the elastic-plastic strain, the load–notch-strain curve for the initial load can be obtained, as described in Section 2.2. Analogous to this procedure, the load–notch-strain curve for the hysteresis branch is determined; however, the double initial load curve is used as the material law (Equation (6)). The number of load steps corresponds to twice the number of classes.

In total, the number of load steps that need to be calculated using FEA to determine the load–notch-strain curves for the initial load and hysteresis branch corresponds to three times the number of classes. In the case of the recommended number of classes of 100, as defined in the FKM guideline nonlinear [25], 300 load steps with elastic-plastic material behaviour need to be calculated.

## 3. Possibilities for the Efficient Design of FEA for the Calculation of the Load–Notch-Strain Curve

An FEA with elastic-plastic material behaviour requires a significantly longer calculation time compared to an FEA with purely elastic material behaviour. The calculation of the load–notch-strain curves via the method described in Section 2.3 is therefore associated with considerable effort, which makes the application appear less attractive compared to the notch root approximation. This section therefore describes the following influences on the quality and efficiency of the simulative determination of load–notch-strain curves:

- The relationship between the load–notch-strain curve for the initial load and the hysteresis branch;
- The definition of the material law in the FE software;
- The optimisation of the FE meshing through the calculation of nodal stresses;
- The number of load steps to be simulated and interpolation.

Some of the listed aspects provide several possible variants for the calculation. These are subjected to a quality and efficiency evaluation in Section 4 in an attempt to give the user a recommendation for the most efficient and, at the same time, accurate calculation of the load–notch-strain curve.

### 3.1. Relationship between the Load–Notch-Strain Curve for the Initial Load and the Hysteresis Branch

Between the load–notch-strain curve for the initial load and the load–notch-strain curve for the hysteresis branch, there exists a relationship by which the load–notch-strain curve of the initial load can be inferred from the load–notch-strain curve of the hysteresis branch without simulation. By halving the ranges of the elasticity-theoretical stress $\Delta\sigma_{el}$ and the strain ranges $\Delta\varepsilon$ of every second class of the load–notch-strain curve of the hysteresis branch, the load–notch-strain curve of the initial load is obtained [48]; see Equations (9) and (10), where $n_C$ is the number of classes. Figure 5 illustrates this procedure, following the descriptions given in Section 2.2. Due to this connection, it is only necessary to carry out the FEA with the double stress–strain curve according to Equation (6) for the hysteresis branch; the load–notch-strain curve for the initial load can be calculated from this. Thus, one-third of the load steps to be simulated are unnecessary.

$$\sigma_{el,i} = \frac{\Delta\sigma_{el,2\cdot i}}{2} \qquad \text{with } i = 1, \ldots, n_C \tag{9}$$

$$\varepsilon_i = \frac{\Delta\varepsilon_i}{2} \qquad \text{with } i = 1, \ldots, n_C \tag{10}$$

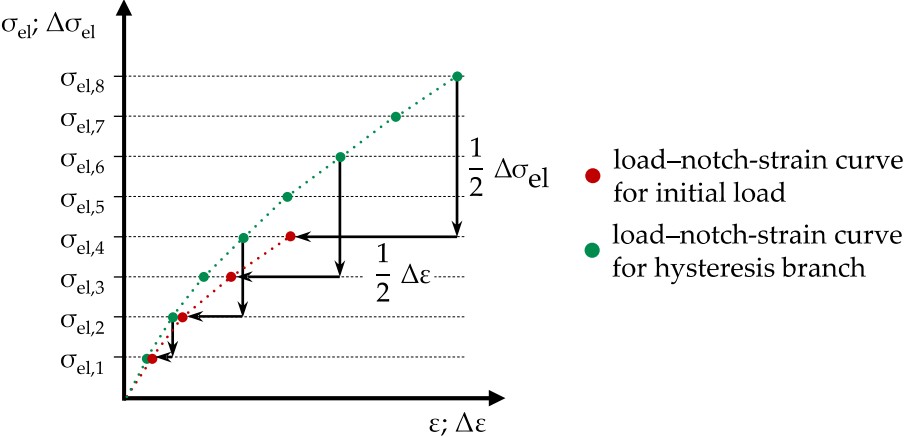

**Figure 5.** Relationships between the load–notch-strain curves for the initial load and the hysteresis branch.

### 3.2. Definition of the Material Law in the FE Software

As described in Section 3.1, it is sufficient to determine the load–notch-strain curve for the hysteresis branch by simulation. For the material in the FE model, the material behaviour according to Equation (6) must be stored as nonlinear hardening. Since only single-sided increasing stresses are simulated in the FEA, it is irrelevant whether kinematic or isotropic hardening is used, since in both cases the same stress–strain path results up to the first load reversal point [69]. In commercial FE programs (such as the Ansys Workbench software used in the following), the material law is often not specified as a function, but as a path of linear sections. The number of supporting points of the material law $n_M$ has an effect on the calculation time and accuracy of the load–notch-strain curve.

The material law is specified in the FE programme up to a stress range of twice the tensile strength $\Delta\sigma = 2\cdot R_m$, as this normally covers all stress ranges that occur. At the end of the FEA, whether the limits of the material law have been exceeded must be checked and, if necessary, the material law must be adjusted. The stresses and corresponding strains are specified with constant spacing in the stress direction.

The required calculation time and the accuracy with which the material law is represented in comparison to Equation (6) depend on the number of support points $n_M$ (index M for material behaviour). The more support points are used, the more accurately Equation (6) is approximated, but the computational effort is also expected to increase.

$$\Delta\sigma_k = \frac{2 \cdot R_m}{n_M} \cdot k \qquad \text{with } k = 1, \ldots, n_M \qquad (11)$$

$$\Delta\varepsilon_k = \frac{\Delta\sigma_k}{E} + 2\left(\frac{\Delta\sigma_k}{2 \cdot K'}\right)^{\frac{1}{n'}} \qquad \text{with } k = 1, \ldots, n_M \qquad (12)$$

### 3.3. Optimisation of the FE Meshing

Another approach to increase the efficiency of determining the load–notch-strain curve is to optimise the meshing of the FE model. The finer the mesh in the area of the point of interest in the component is, the more accurately the stresses and strains can be determined. In contrast, the simulation time increases with finer meshing.

By meshing the component, it is divided into discrete sections—the so-called finite elements. These finite elements consist of element nodes, which lie at the edge of the elements, and integration points, also called Gauss points, which—as a rule—lie inside the elements. The integration points are the positions in the element at which the result values determined by integration are valid. In Figure 6A, a hexahedral serendipity solid element with quadratic shape function is shown schematically. The Gauss points are marked in red and the element nodes in green. Since the result variables at the element nodes are usually of interest, these are calculated from the results at the Gauss points with interpolation or extrapolation [70]. Only this type of element (Ansys Solid 186 [71]) is used for the investigation in Sections 4 and 5 in the area of the surface to be evaluated. However, to make sure that other common element types lead to qualitatively similar results, the results are checked using the following other element types as examples:

- Hexahedron elements with linear shape function and full integration (Ansys Solid 185 [72]);
- Tetrahedron elements with linear shape function and full integration (Ansys Solid 185 [72]);
- Tetrahedron elements with quadratic shape function and reduced integration (Ansys Solid 187 [73]).

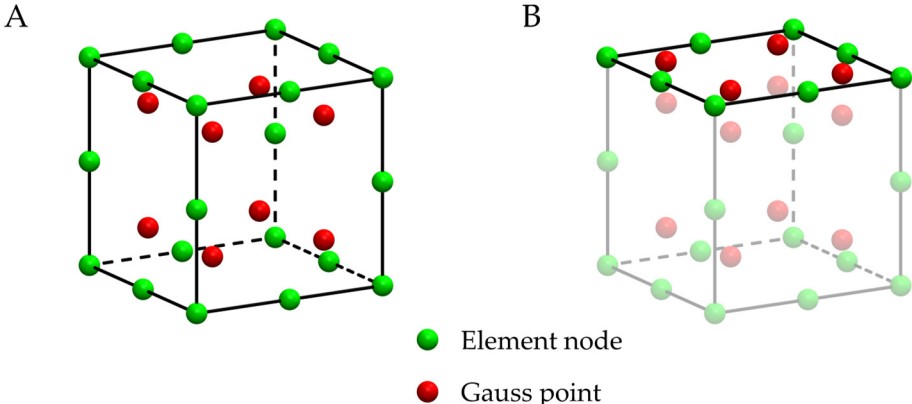

**Figure 6.** Gauss points and element nodes: (**A**) hexahedron element (Ansys Solid186); (**B**) hexahedron element with shell element (Ansys Shell281) on one surface.

The exemplary investigation shows: with appropriate mesh quality, all types of elements mentioned above lead to qualitatively the same results. Fully integrated elements with a quadratic shape function, however, are not recommended due to the investigations in [74].

This procedure is common in Ansys Workbench when using linear material behaviour, and the extrapolation provides a good estimate of the results gained at the element nodes. In the case of nonlinear material behaviour, the extrapolation of the results to the element

nodes can lead to large errors [70]. These errors become larger the greater the distance between the integration point and the element node is; i.e., if the element is larger, this is also the error that occurs. Therefore, when plastic deformation occurs in the element, often no extrapolation is carried out; instead, the results gained at the integration points are copied to the element nodes [70]. This leads to an underestimation of the stresses and strains on the surface of the component and thus to an overestimation of the service life.

The strain gradient in the notch root is relatively large in the direction vertical to the component surface. It is possible to determine the stresses and strains at the element nodes as accurately as possible if the Gauss points are as close as possible to the element nodes in the area of the component surface. On the one hand, this can be achieved by using a finer mesh at the expense of computing time. On the other hand, shell elements can be applied to the solid elements of the model at the surface of the component if only the stresses at the component's surface are of interest. Since shell elements are 2D elements, the Gauss points are on the surface; see Figure 6B. This is expected to reduce the error caused by copying or extrapolating the results and to converge the result variables to a fixed value, even when a coarser mesh is used.

### 3.4. Reduction in the Number of Load Steps to Be Simulated

Another way to reduce the simulation time is to decrease the number of load steps to determine the load–notch-strain curve for the hysteresis branch. If the number of simulated load steps is reduced, not every class a value for the load–notch-strain curve is calculated. In order to still be able to use the same algorithm for the fatigue life calculation with the same number of discrete load classes, the missing intermediate class values must be determined by interpolation or regression of the calculated load steps. For the interpolation or regression, three procedures are presented in the following:

1.  Interpolation by polygonal chain (polynomials of 1st degree);
2.  Regression with power function following the approach of Ramberg and Osgood (Equation (3));
3.  Interpolation with spline (polynomials of 3rd degree).

All three methods have in common the relationship $\Delta\varepsilon(\Delta\sigma_{el})$ between the elastic-plastic strain range and the elastic stress range results, which is used to describe the load–notch-strain curve. The discretised load–notch-strain curve for the hysteresis branch is obtained by inserting the elastic stress ranges at the class limits in this relationship.

### 3.4.1. Linear Interpolation by Polygonal Chain

A simple possibility for interpolation between the calculated results is to form a polygonal chain of 1st-degree polynomials. For the number of load steps $n_H$ (with index H for the hysteresis branch) calculated to determine the load–notch-strain curve, $n_{H+1}$ support points result for the linear interpolation, since in addition to the support points $\Delta\varepsilon_{FEA,2} \dots \Delta\varepsilon_{FEA,nH+1}$ at $\Delta\sigma_{el,FEA,2} \dots \Delta\sigma_{el,FEA,nH+1}$ from the FEA, point $\Delta\varepsilon_{FEA,1} = 0$ at $\Delta\sigma_{el,FEA,1} = 0$ is available as a support point. The polygonal chain consists of $n_H$ sections to calculate the strain range $\Delta\varepsilon_L$ (index L for linear interpolation), which is defined in a sectionwise manner; see Equation (13). Slopes $a_i$ and offsets $b_i$ can be calculated directly from the FEA results using Equations (14) and (15).

$$\Delta\varepsilon_{L,i}(\Delta\sigma_{el}) = a_i \cdot (\Delta\sigma_{el} - \Delta\sigma_{el,FEA,i}) + b_i \quad \begin{array}{c} \text{if } \Delta\sigma_{el,FEA,i} \le \Delta\sigma_{el} \le \Delta\sigma_{el,FEA,i+1} \\ \text{with } i = 1, \dots, n_H \end{array} \tag{13}$$

$$a_i = \frac{\Delta\varepsilon_{FEA,i+1} - \Delta\varepsilon_{FEA,i}}{\Delta\sigma_{el,FEA,i+1} - \Delta\sigma_{el,FEA,i}} \tag{14}$$

$$b_i = \Delta\varepsilon_{FEA,i} \tag{15}$$

Figure 7 shows the procedure for two simulated load steps $n_H = 2$. As a result of the FEA, the strains $\Delta\varepsilon_{FEA,2}$ and $\Delta\varepsilon_{FEA,3}$ are obtained for the applied loads $\Delta\sigma_{el,FEA,2}$ and $\Delta\sigma_{el,FEA,3}$. The values for the load–notch-strain curve at the class limits are taken from the interpolated red curve by inserting the elasticity-theoretical stresses $\Delta\sigma_{el}$ at the class limits into Equation (13). As the number of simulated loads increases, the interpolated load–notch-strain curve approaches the true curve.

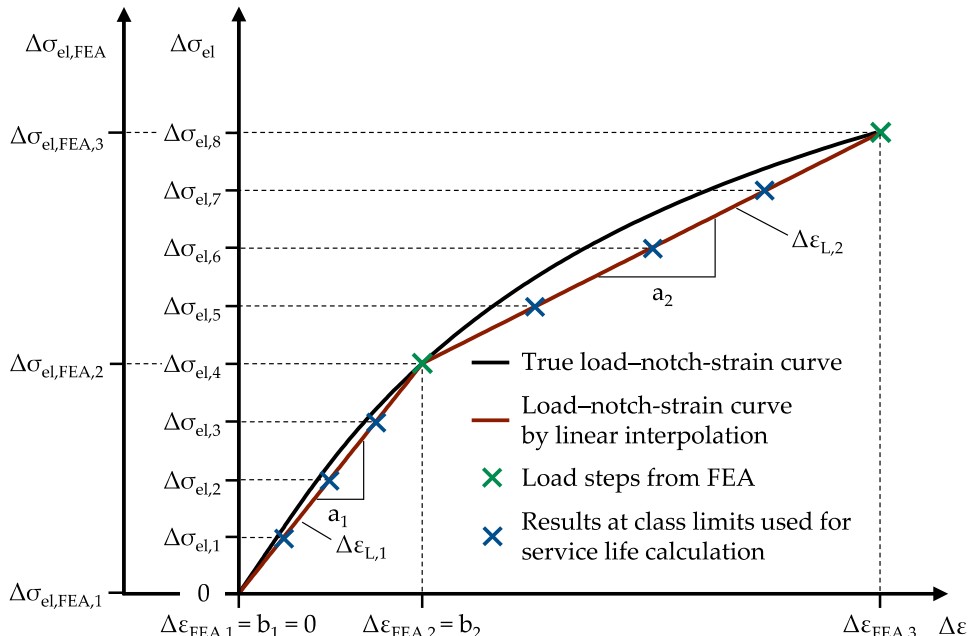

**Figure 7.** Interpolation of the load–notch-strain curve by polygonal chain with $n_H = 2$ load steps.

3.4.2. Regression with Power Function

As an alternative to the interpolation with a polygonal chain, the load–notch-strain curve can be described by a power function. The power function is chosen based on the material law according to Ramberg and Osgood [44]; see Equation (3). Ball [15] also used this approach, and was thus able to describe load–notch-strain curves from FEA with a functional relationship in a good approximation. As described in Section 3.2, only the determination of the load–notch-strain curve for the hysteresis branch is necessary. This is described with Equation (16) following Equation (6), with which the range of the strain $\Delta\varepsilon_P$ (index P for power function) is calculated:

$$\Delta\varepsilon_P(\Delta\sigma_{el}) = \frac{\Delta\sigma_{el}}{E} + 2\left(\frac{\Delta\sigma_{el}}{2\cdot c}\right)^{\frac{1}{d}} \tag{16}$$

The aim is now to determine the parameters c and d. For this purpose, the front part—Hooke's law described by the modulus of elasticity E—is subtracted from the stress range $\Delta\varepsilon_P$. The remaining power approach represents a linear course by double logarithmic scaling. A comparison with the results from the FEA in Figure 8 obviously does not show a linear course. This can be explained by the approximation to the net section plasticity [28,75].

A linear regression through all the simulated load steps results in the red curve, which shows deviations from the simulated ones in all areas. In particular, the load steps for small loads $\Delta\sigma_{el}$ deviate strongly from a linear course; thus, it makes sense to exclude them for a regression. If only load steps above a proposed limit value of $\log(\Delta\varepsilon_{FEA} - \Delta\sigma_{el,FEA}/E) \geq -4$ are considered in the regression, the load steps of the medium and high loads can be well approximated (see the black curve). The large deviation found for small loads only appears very large due to the logarithmic scaling applied.

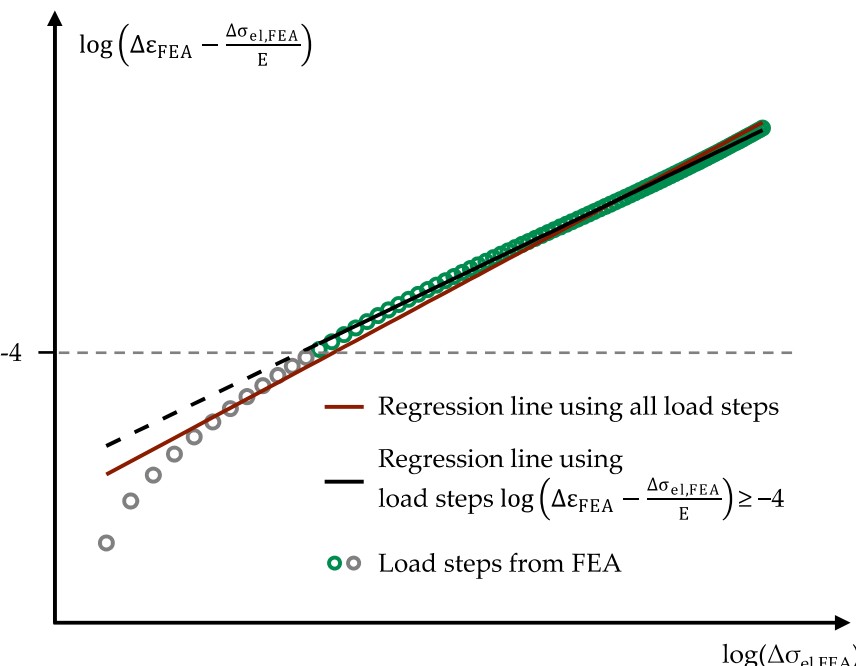

**Figure 8.** Regression of the load–notch-strain curve with the power approach.

Figure 9 shows the load–notch-strain curves with both regressions in a linear representation. In the range of low loads, there is no apparent difference between the load–notch-strain curves; in the range of high loads, the load–notch-strain curve with consideration of the limit value of $\log(\Delta\varepsilon_{FEA}-\Delta\sigma_{el,FEA}/E) \geq -4$ reproduces the results of the FEA significantly better. Therefore, this limit value was used in the following. More details on the regression and the determination of the parameters c and d are given in Appendix A.

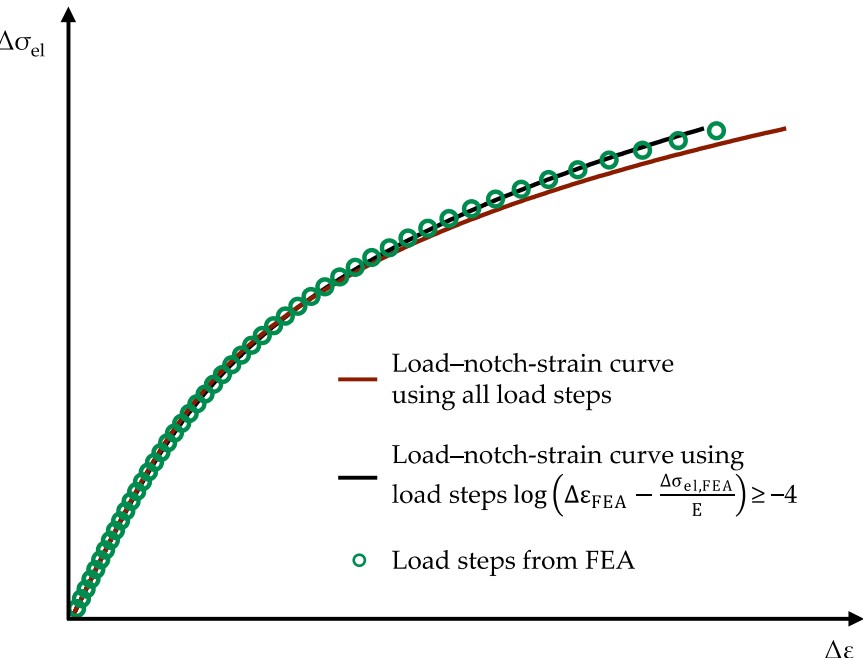

**Figure 9.** Load–notch-strain curve with the power approach.

The stress ranges at the class limits for the load–notch-strain curve of the hysteresis branch are then determined using Equation (16). For the load–notch-strain curve for the initial loading, proceed as in Section 3.2 or use Equation (17):

$$\varepsilon_P(\sigma_{el}) = \frac{\sigma_{el}}{E} + \left(\frac{\sigma_{el}}{c}\right)^{\frac{1}{d}} \tag{17}$$

The advantage of this approach is that the load–notch-strain curve is not described by a sectionwise-defined function. This makes it possible, for example, to estimate load–notch-strain curves from the existing FEA results when the load requirements change, even if the new maximum load $L_{max}$ is slightly higher than the maximum simulated load. This is not readily possible with sectionwise-defined functions. The disadvantage of this is that the load–notch-strain curve does not necessarily run through the FEA results due to the linear regression.

### 3.4.3. Interpolation by Spline

Similar to interpolation with 1st-degree polynomials, sectionwise-defined 3rd-degree polynomials are used in the spline interpolation; see Equation (18):

$$\Delta\varepsilon_{S,i}(\Delta\sigma_{el}) = a_i \cdot (\Delta\sigma_{el} - \Delta\sigma_{el,FEA,i})^3 + b_i \cdot (\Delta\sigma_{el} - \Delta\sigma_{el,FEA,i})^2 + c_i \cdot (\Delta\sigma_{el} - \Delta\sigma_{el,FEA,i}) + d_i$$
$$\text{if } \Delta\sigma_{el,FEA,i} \leq \Delta\sigma_{el} \leq \Delta\sigma_{el,FEA,i+1}$$
$$\text{with } i = 1, \dots, n_H \tag{18}$$

A boundary condition for determining the parameters $a_i$, $b_i$, $c_i$, and $d_i$ is, among other things, the slope at the origin of the load–notch-strain curve with the modulus of elasticity E. Details of the determination of the parameters are given in Appendix B. Despite using this boundary condition, in cases where there are very few load steps $n_H$ from FEA, the case may be that the strains calculated through spline interpolation are smaller than the linear-elastic strains. Although this is mathematically possible through spline interpolation, it does not make physical sense. For this reason, Equation (19) is used:

$$\Delta\varepsilon_S(\Delta\sigma_{el}) = \begin{cases} \frac{\Delta\sigma_{el}}{E} & \text{if } \Delta\varepsilon_S < \frac{\Delta\sigma_{el}}{E} \\ \Delta\varepsilon_S & \text{if } \Delta\varepsilon_S \geq \frac{\Delta\sigma_{el}}{E} \end{cases} \tag{19}$$

Figure 10 shows a schematic load–notch-strain curve derived from two simulated load steps using spline interpolation. This is compared with the true load–notch-strain curve. The values for the load–notch-strain curve at the class limits are taken from the interpolated red curve by inserting the elasticity-theoretical stresses $\Delta\sigma_{el}$ at the class limits into Equation (18).

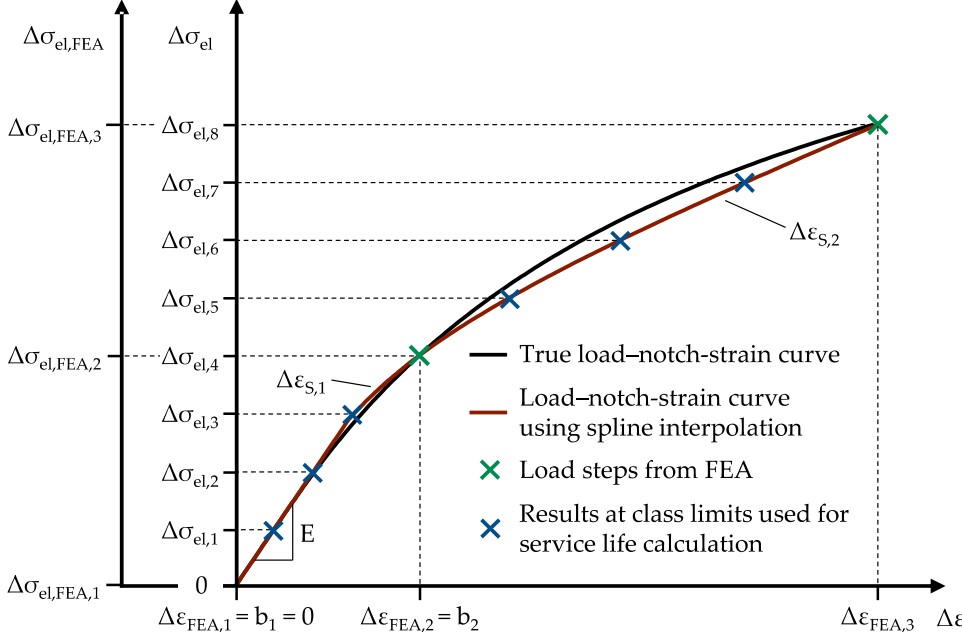

**Figure 10.** Load–notch-strain curve with spline interpolation from two load steps ($n_H$ = 2).

## 4. Evaluation of Quality and Efficiency

In this section, the previously described methods for determining load–notch-strain curves with the help of FEA were examined to determine their quality and efficiency. For this purpose, quality and efficiency criteria were first defined, and a database with different component geometries, load types, and material parameters was created. Then, using the methods presented in Section 3, load–notch-strain curves were generated for the components from the database and evaluated with the quality and efficiency criteria.

### 4.1. Quality and Efficiency Criteria

In order to evaluate the methods described, the load–notch-strain curves generated with the methods described in Section 3 are compared to a reference load–notch-strain curve. The reference load–notch-strain curve is regarded as the "true" load–notch-strain curve and is determined by FEA with a very fine mesh.

Alternatively, experimentally determined load–notch-strain curves or experimentally determined fatigue lives could also be used as a reference. However, this has the decisive disadvantage that certain simplifying assumptions made by the FKM guideline nonlinear cannot be exactly reproduced by real components or specimens. These include, for example, the description of the material behaviour by the approach according to Ramberg and Osgood or the assumption of a cyclically stabilized material behaviour. Furthermore, the comparison of calculated and experimental results is expected to show a bias in most cases. This bias, however, is due to the calculation concept itself and the accuracy of the FKM guideline nonlinear, and not the procedure for estimating the load–notch-stain-curve that shall be examined in this contribution.

The approximated load–notch-strain curves generated using the methods described in Section 3 aim to reproduce the true curve as accurately as possible. Since the simplified generated load–notch-strain curve and the reference curve each consist of pairs of values from elastic stress and notch strain, and the associated local stress is also used as an input value for service life calculation in addition to the notch strain, a quality evaluation based on the load–notch-strain curve alone does not lead to meaningful results as a basis for a recommendation. The more interesting result for a user is the effect on the calculated service life. Therefore, the latter is used for the quality evaluation. For the service life calculation, the previously mentioned algorithms of the FKM guideline nonlinear [25] are used. For the load–notch-strain curve, the curves determined with FEA are used. The calculation procedure—reduced to the most necessary steps—is shown schematically in Figure 11 and explained briefly below.

On the one hand, information about the component, such as the geometry and tensile strength of the material, are required as input variables. On the other hand, the load-time history and load configuration, meaning the vector of all forces and/or moments, must be known. The starting point for describing the material behaviour is the tensile strength. From this, the corresponding cyclic material properties $K'$ and $n'$ are estimated with the method according to Wächter et al. [45,46]; see Equations (4) and (5). The quality ratings in this contribution are limited to estimated cyclic properties for the material group steel. However, a significant deviation in the findings determined in this manner to other material groups is not expected. With the cyclic material properties estimated in this way, the material law can be inputted into the FEA to determine the load–notch-strain curve. The geometry is then transferred to the FEA, loads are applied according to the specified load configuration, and the load–notch-strain curve is determined as described above, considering the maximum load occurring in the load-time history.

In addition, the load parameter Wöhler curve is estimated from the tensile strength, which also takes into account the influences of geometry and load type by considering support factors due to stress gradients and size effects. Thus, the service life is determined for a failure probability of 50%. Since the service lives are only used for a qualitative comparison, the application of the safety concept contained in the FKM guideline nonlinear is not necessary. The safety factors are set to 1.

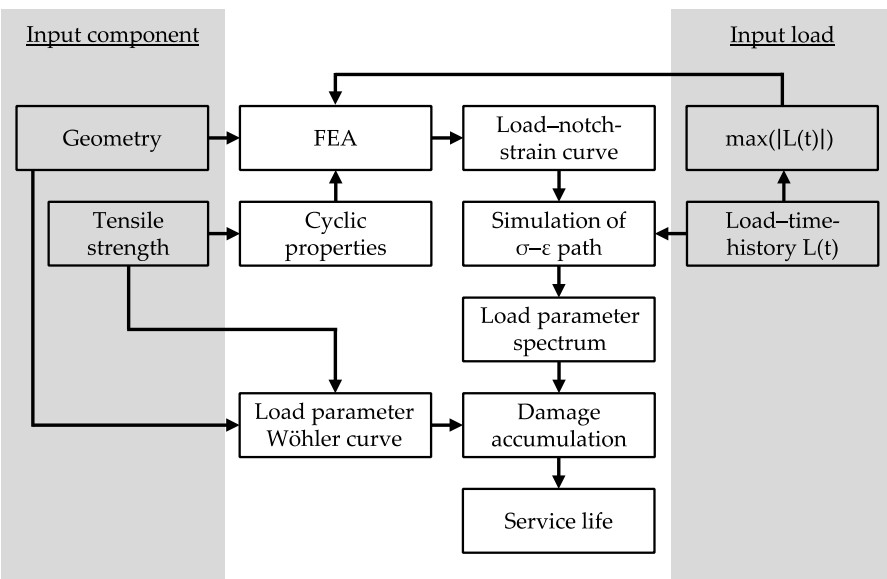

**Figure 11.** Procedure used for service life estimation based on the FKM guideline nonlinear [25].

Another input variable is the load on the component. Two load-time histories are used in this paper to compare the service lives:

- A load sequence with a constant amplitude;
- A normally distributed load sequence with irregularity factor I = 0.7 [76].

Figure 12A shows the load sequence with a constant amplitude, and Figure 12B shows a section of 0.01% of the reversal points of the normally distributed load sequence [76], which contains the maximum load. By using a constant amplitude loading and a variable amplitude loading, the two most relevant load cases are considered in the quality rating.

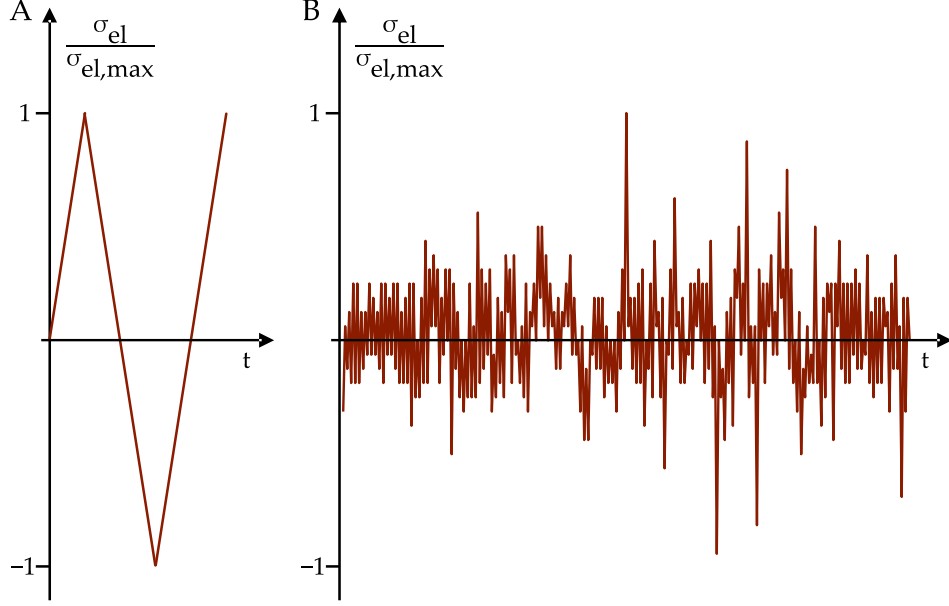

**Figure 12.** Load-time history: (**A**) constant amplitude loading; (**B**) 0.01% of the normally distributed load sequence with I = 0.7 [76].

From the load–notch-strain curve and the load-time history, the local stress–strain paths are simulated using the principles described in Section 2.2, and the closed hystereses are determined. For each closed hysteresis, a load parameter $P_{RAM}$ (see Equation (1)) is cal-

culated, and a load parameter spectrum is created. The second possible damage parameter $P_{RAJ}$ [25] (see Equation (2)) is not considered for carrying out the quality assessment. Since this is a comparison of two calculated service lives that differ only in the load–notch-strain curve used, no qualitatively different results are expected if $P_{RAJ}$ is used.

Using linear damage accumulation, a service life is calculated from the load parameter spectrum and the load parameter Wöhler curve. For the quality assessment, this procedure is carried out once with a reference load–notch-strain curve and once with load–notch-strain curve using the methods described in Section 3. Afterwards, the resulting service lives are put into relation. The boundary conditions used for determining the reference load–notch-strain curve and the load–notch-strain curve with the methods from Section 3 are given in each section of the quality assessment.

The efficiency criterion is the comparison of the sum of all computation times on all computation kernels used to determine the reference load–notch-strain curve or the load–notch-strain curve with the methods given in Section 3. The sum of all the computing times is referred to as CP-Time (computing time).

*4.2. Database*

An extensive database is created for a quality and efficiency evaluation. For this purpose, load–notch-strain curves are determined for different geometries with different load configurations and notch geometries. These geometries can be classified as follows, see Figure 13:

- Flat specimen with notches on both sides loaded with tension or bending, and stress concentration factors $K_t$ = 1.5, 3, and 5;
- Circular specimen with notches loaded with tension, bending, or torsion; and stress concentration factors $K_t$ = 1.5, 3, and 5;
- Planetary carrier loaded with planetary pin forces.

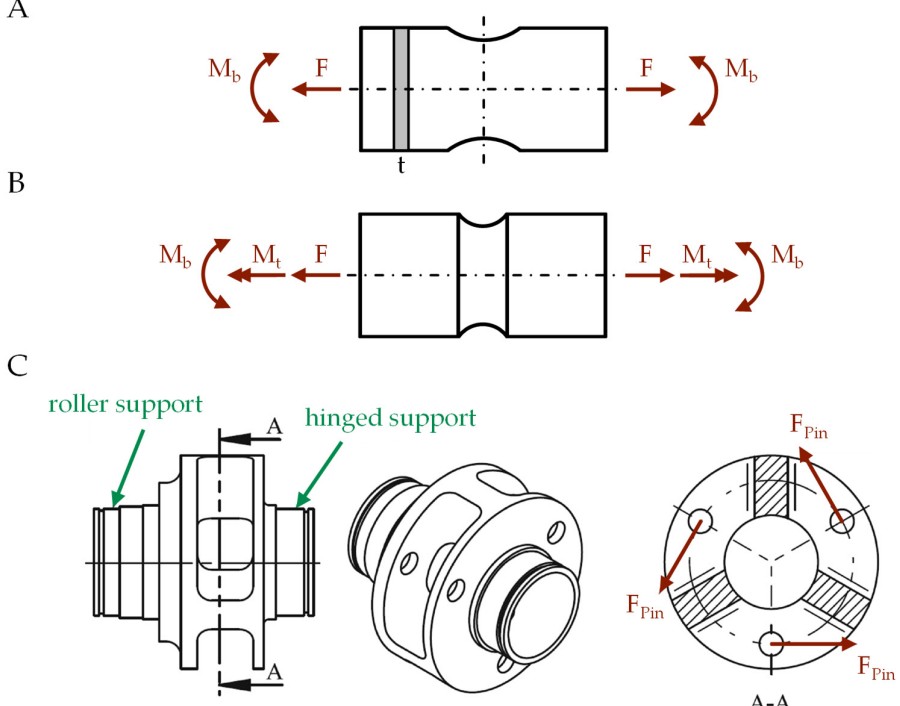

**Figure 13.** Exemplary specimen geometries and load types of the database: (**A**): flat specimen with $K_{t,tension}$ = 1.5; (**B**) round specimen with $K_{t,tension}$ = 1.5; (**C**) planetary carrier from [69].

In the investigations detailed in the following sections, different parts of the database are utilised.

As mentioned before, the investigations are limited to the material group of steel. In the FEA, E = 206 GPa is used as the modulus of elasticity. For the tensile strengths, $R_m$ = 400 MPa, 800 MPa, and 1200 MPa are chosen. The first two values are taken as relevant lower and average strengths for steels, while the upper value corresponds to the upper limit of the range of validity of the FKM guideline nonlinear [25]. As described in Section 4.1, the cyclic properties are estimated from the tensile strength using Equations (4) and (5).

The load sequences shown in Figure 12 are varied by using different maximum values of the load spectrum. The height of the maximum values is chosen in such a way that approximately the same service life results in a constant amplitude loading $N_{const.}$ for different geometries. In order to compare the data for the different geometries, the loads L are expressed in the form of linear-elastic stresses in the following. Table 1 summarises the number of cycles with the corresponding symbols for the stress range of the linear-elastic stress. For the high load $L_{max,3}$, the stress range at constant amplitude loading is $\Delta\sigma_{el,max,3}$, which, under the previously described boundary conditions, results in a calculated service life of $N_{const.} \approx 10^2$ cycles. To illustrate the loads, they are plotted on an exemplary load–notch-strain curve; see Figure 14. At the low load $L_{max,1}$, the stress range of the theoretical elastic stress $\Delta\sigma_{el,max,1}$ features a large elastic strain part and a small plastic strain part, whereas at the high load $L_{max,3}$, the plastic strain part dominates.

**Table 1.** Examined maximum values of the load spectra.

| Maximum Value of the Spectrum $L_{max}$ | Linear-Elastic Stress Range | Cycles $N_{const.}$ |
|---|---|---|
| Low load $L_{max,1}$ | $\Delta\sigma_{el,max,1}$ | $\approx 10^5$ |
| Medium load $L_{max,2}$ | $\Delta\sigma_{el,max,2}$ | $\approx 10^3$ |
| High load $L_{max,3}$ | $\Delta\sigma_{el,max,3}$ | $\approx 10^2$ |

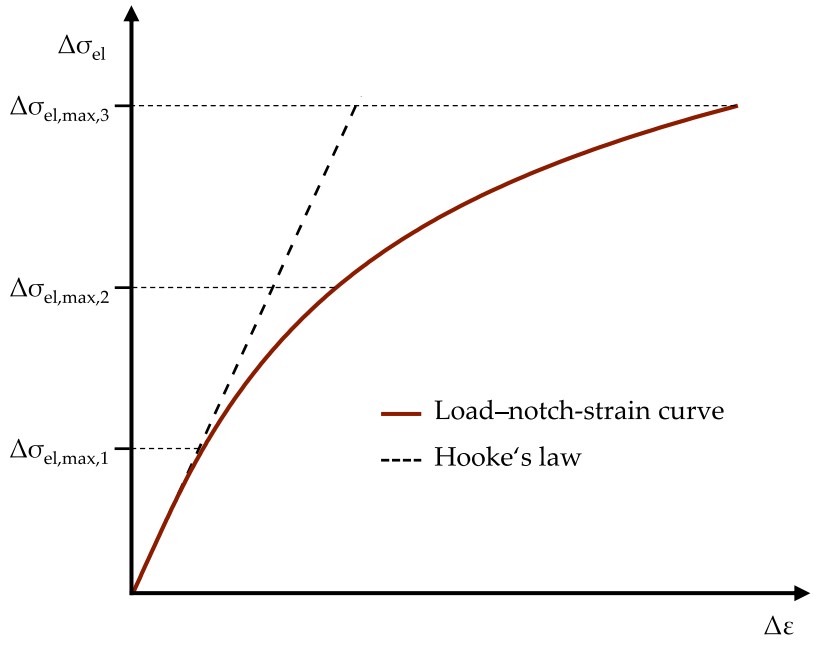

**Figure 14.** Maximum values of load spectra illustrated on a load–notch-strain curve.

The load steps are applied in the simulation in equidistant steps so that the last load step corresponds to the maximum load $L_{max}$. The load steps $L_{I,k}$ used for determining the load–notch-strain curve for the initial load are based on Equation (20) with the number of load steps $n_I$ (index I for initial load). Similarly, the load steps $L_{H,k}$ used for determining the load–notch-strain curve for the hysteresis branch are based on Equation (21) with the number of load steps $n_H$. For the specifications of the FKM guideline nonlinear, $n_I = 100$

and $n_H = 200$ are used, resulting in a total of 300 load steps that need to be calculated for the reference load–notch-strain curve:

$$L_{I,k} = \frac{L_{max}}{n_I} \cdot k \qquad \text{with } k = 1, \ldots, n_I \qquad (20)$$

$$L_{H,k} = \frac{2 \cdot L_{max}}{n_H} \cdot k \qquad \text{with } k = 1, \ldots, n_H \qquad (21)$$

### 4.3. Relationship between the Load–Notch-Strain Curve for the Initial Load and the Hysteresis Branch

Due to the relationship between the load–notch-strain curve for the initial load and the hysteresis branch, it is not necessary to determine the load–notch-strain curve for the initial load; see Section 3.1. This does not result in any loss of quality, but the efficiency can be increased because the number of load steps that need to be calculated is reduced by one-third. In the following sections, this insight is used, and only the determination of the load–notch-strain curve for the hysteresis branch is discussed.

### 4.4. Definition of the Material Law in the FE Software

In order to determine the influence of the number of support points $n_M$ for the material law on the load–notch-strain curve, load–notch-strain curves are generated on a flat specimen notched on both sides while varying the number of support points in the material law. The flat specimen with a stress concentration factor $K_t = 3$ and a tensile strength $R_m = 800$ MPa is loaded in tension; see Section 4.2. The reference load–notch-strain curve is determined with the same boundary conditions and $n_M = 200$ support points for the material law.

The service lives for constant amplitude loading $N_{const.,n_M}$ and variable amplitude loading $N_{var.,n_M}$ serve as quality criteria, and the computing $CP - Time_{n_M}$ serves as the efficiency criterion. Figure 15 shows the comparative values related to the value at $n_M = 200$ support points for a low load $L_{max,1}$ and a high load $L_{max,3}$.

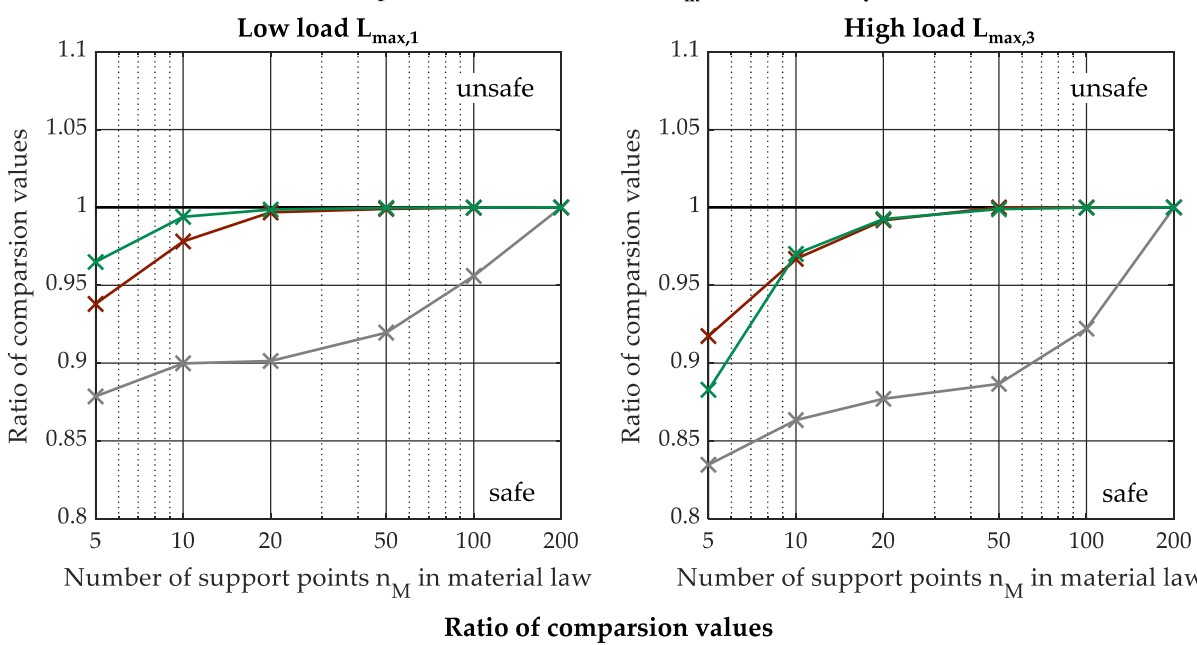

**Figure 15.** Influence of the number of support points $n_M$ of the material law on the load–notch-strain curve.

Both the service life at constant and variable amplitude loading are underestimated for only few support points $n_M$, regardless of the load level, resulting in a conservative service life estimation in the applied case. At low loads, the service life converges more quickly to a constant value than it does at high loads, where no influence of the number of support points on the service life can be seen from $n_M = 50$ support points on. The greater the number of support points is, the longer the simulation time is. In the case considered here, this results in time savings of about 10% when using $n_M = 50$ support points compared to $n_M = 200$ support points.

In Figure 16, the influence of the number of support points $n_M$ for the material law on the service life with variable amplitude loading is examined for all flat specimens of the database under tensile loading at high load $L_{max,3}$; see Section 4.2. Here, too, it can be seen that the material law is described precisely enough for 50 and more support points $n_M$ for all examined specimens. It may be concluded that from 50 support points onwards, no significant influence on the service life and thus on the load–notch-strain curve results. With less than 50 support points $n_M$, especially the slightly notched specimens react strongly to the poorly described material law.

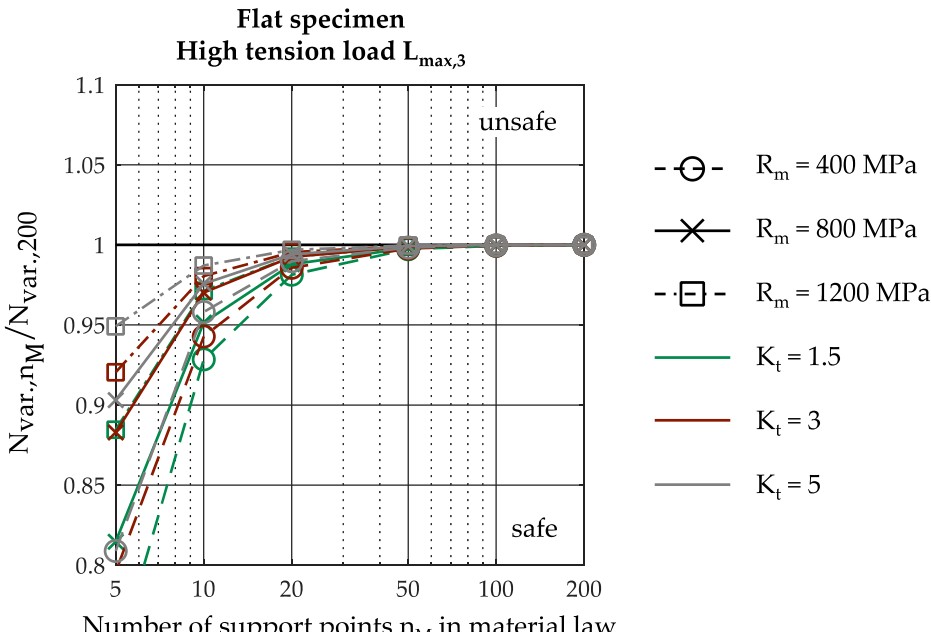

**Figure 16.** Influence of the number of support points $n_M$ of the material law on the load–notch-strain curve at different $R_m$ and $K_t$.

As a recommendation for the user, $n_M = 50$ support points are therefore suggested to describe the material law. In order to exclude the influence of inaccuracies in the material law in the following sections, the value $n_M = 100$ is used below.

### 4.5. Optimisation of the FE Meshing

The influence of the mesh fineness on the quality and efficiency was determined, considering:

- Copying the results from the Gauss points to element nodes;
- The extrapolation from the Gauss points to element nodes;
- The use of additional shell elements on the surface.

These are investigated on a flat specimen under axial load—see Section 4.2. The specimen consists of steel with a tensile strength of $R_m = 800$ MPa and a stress concentration factor of $K_t = 3$. When extrapolating the result variables to the element nodes, the largest errors occur with large plastic deformations. Therefore, a single load is applied with the

high load $L_{max,3}$ described in Section 4.2. The meshing of the specimen is conducted exclusively with hexahedron elements with a quadratic element shape function (Ansys Solid186 [71]). Other element types are also suitable for meshing the component, but these are not considered here. The user is advised to check the quality of the mesh themselves in any case in order to obtain converged stresses and strains and to exclude singularities. The meshing in the notch root is described by the number of elements $n_E$ over 90°; see Figure 17. For the elements in the notch root, edge lengths of almost the same length are used, meaning that a change in the number of elements $n_E$ over 90° also leads to a change in the number of elements in the other directions. In addition, shell elements are added in the notch root at the component surface. In Ansys Workbench, the addition of shell elements made possible through the surface coating function [77]. Shell elements with a linear element shape function (Ansys Shell181 [78]) or with a quadratic element shape function (Ansys Shell281 [79]) are used according to the element shape function of the solid elements used.

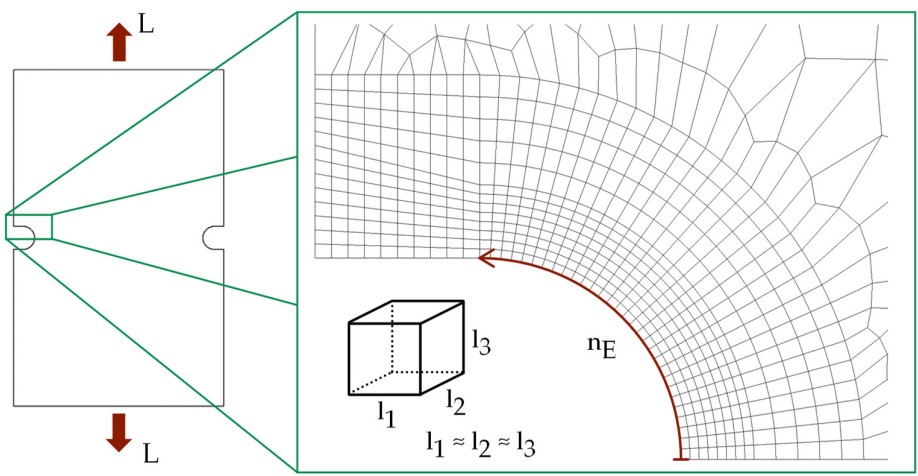

$n_E$: Number of Elements per 90°

**Figure 17.** Meshing in the notch root of the flat specimen with $K_t = 3$ and $n_E = 30$.

Figure 18 shows the influence of the mesh and the evaluation method used on the strain range $\Delta\varepsilon$ and the service life at constant amplitude loading $N_{const.}$. It can be observed that with increasingly finer meshing, the determined load–notch-strain curve and thus the calculated fatigue life asymptotically approach a limit value. This limit value is to serve as a reference for the evaluation of the optimizations shown below in the determination of the load–notch-strain curve. Due to the asymptotic approach to the limit value, no practice-relevant difference between the load–notch-strain curves or calculated fatigue lives can be determined above a certain mesh fineness. This mesh quality is reached at the number of elements per 90° $n_E = 70$, and will be used as a reference in the following. The influence of the meshing and the evaluation method in Figure 18A can be summarised as follows:

- In *blue*, strain ranges are shown, which are determined by copying the result variables from the Gauss points to the element nodes. The strain ranges are underestimated, but with finer meshes, and with the associated shift of the Gauss points in the direction of the surface of the area to be analysed, they approach a constant value. This behaviour occurs both when assuming a purely elastic (*dashed line*) and an elastic-plastic (*solid line*) material behaviour.
- When using the extrapolation approach (*red line*), the strain range $\Delta\varepsilon_{el}$ converges towards a fixed value with a significantly coarser mesh. In the case of elastic-plastic material behaviour, after extrapolation in Ansys Workbench, the relationship between stress and strain no longer agrees with the material law according to Equation (6). If the strain ranges from the FEA are used, the curve is almost the same as the strain ranges

determined by copying the result variables from the Gauss points to the element nodes (*blue curve*). The extrapolation has no significant influence. Alternatively, for comparison the strain ranges can be calculated from the stress from FEA using Equation (6), shown as the red curve. Due to the extrapolation, the elastic-plastic strain range $\Delta\varepsilon$ converges faster than it does for copying, but the strain ranges are clearly overestimated with a coarse mesh.

- By using shell elements on the surface and copying the FEA results (*green line*) the same behaviour occurs for elastic and elastic-plastic material behaviour: the strain ranges converge towards a constant value even with a coarse mesh.
- When using shell elements on the surface and the additional application of extrapolation (*grey line*), the strain range converges more slowly, especially with elastic-plastic material behaviour. With elastic-plastic material behaviour, the strains are calculated from the stress as before.

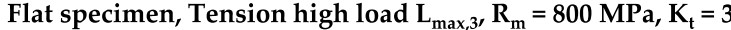

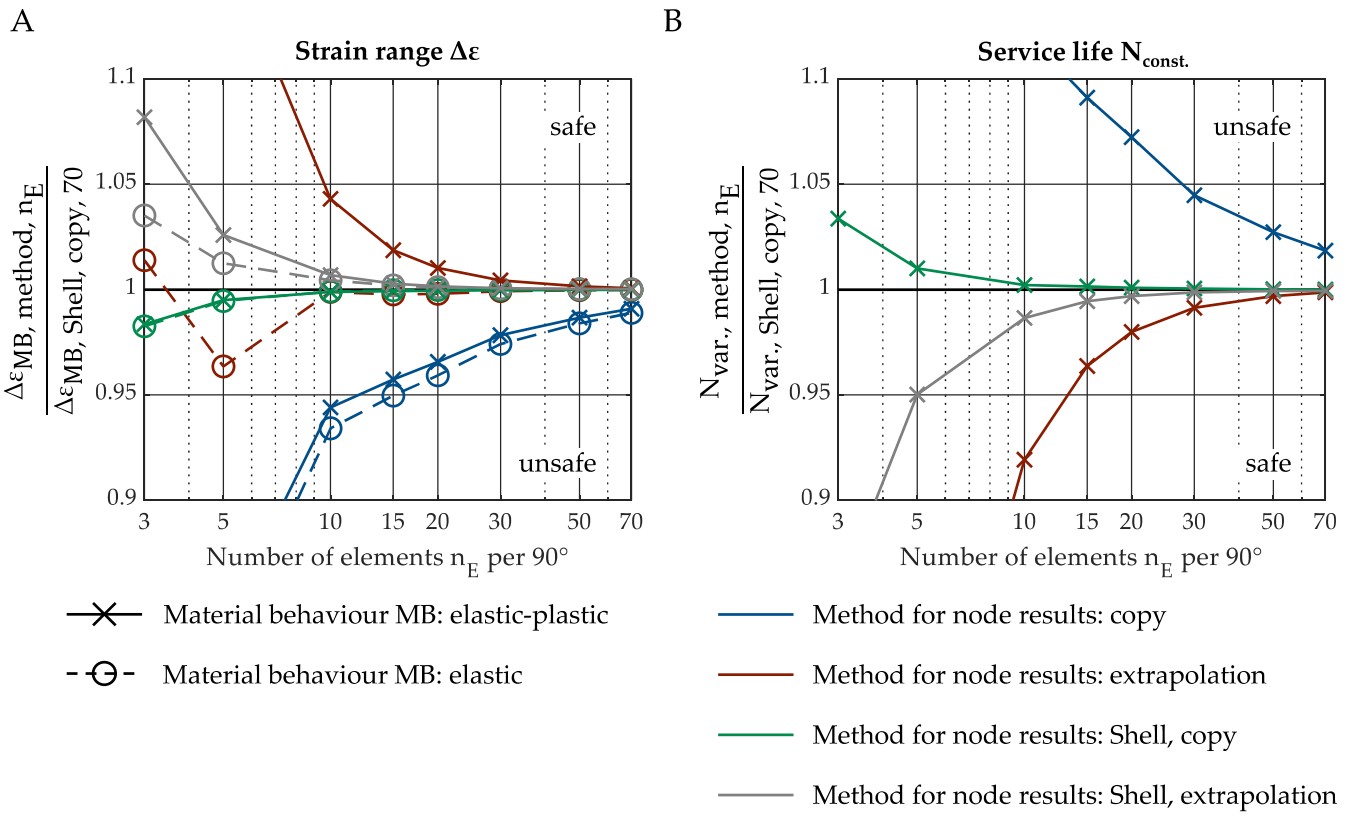

**Figure 18.** Influence of the mesh in elastic and elastic-plastic material behaviour on the: (**A**) strain range; (**B**): service life.

The tendencies shown are also reflected in a consistent manner in the comparison of the service lives at constant amplitude loading in Figure 18B, which are calculated from the results of the simulation with elastic-plastic material behaviour. In the standard procedure in Ansys Workbench involving copying the results at the Gauss points to the element nodes, a mesh with $n_E > 70$ is needed in this case to determine the service life with an error of less than 1%. By using extrapolation, a mesh with $n_E = 30$ is sufficient to fulfil this condition. A great advantage can be gained for the meshing results from the use of shell elements on the surface. Together with extrapolation, a mesh with $n_E = 15$ is sufficient. On the other hand, it is recommended to copy the results at the Gauss points, as this means that a mesh with $n_E = 5$ is already sufficient, even if the service life thereby approaches the converged solution from the unsafe side.

The use of shell elements on the surface along with copying of the results at the integration nodes to the element nodes offers no great advantage in calculations with elastic material behaviour. With elastic-plastic material behaviour, however, a converged result with a significantly coarser mesh can be determined. The resulting time saving increases the efficiency in determining the load–notch-strain curve immensely.

### 4.6. Reduction in the Number of Load Steps to Be Simulated

In this section, the methods for reducing the number of load steps to be simulated from Section 3.4 are compared with each other. For this purpose, load–notch-strain curves for the hysteresis branch with different numbers of load steps $n_H$ = 2, 3, 4, 5, 7, 10, and 20 are determined in the simulation, while the values at the class limits are determined using one of the approaches described in Section 3.4. A reference load–notch-strain curve is calculated with $n_H$ = 200 load steps, whereby the simulation directly calculates the values at the class limits of the load–notch-strain curve for the hysteresis branch.

For the evaluation of the node results, the previously recommended shell elements are used on the surface. However, to be able to exclude deviations resulting from a too-coarse mesh, a finer mesh than the one recommended is used. For example, for the flat sample used in Section 4.5, instead of the recommended number $n_E$ = 5, the number $n_E$ = 20 is used.

However, the results shown in this section are equally valid with and without the use of shell elements on the surface for the evaluation of the node results. In both cases, the meshing is chosen in such a way that converged stresses and strains can be assumed.

#### 4.6.1. Comparison of the Methods on a Notched Flat Specimen

A first comparison of the methods to describe the load–notch-strain curve is carried out on the flat specimen made from steel notched on both sides with a tensile strength of $R_m$ = 800 MPa and a stress concentration factor $K_t$ = 3 under tensile loading.

Figure 19 shows the ratios of the service lives with variable amplitude loading $N_{var.}$ (see Figure 12) from load–notch-strain curves with $n_H$ load steps and service lives from the reference load–notch-strain curve ($n_H$ = 200 load steps). As the number of load steps increases, the service life converges to the service life with 200 load steps. The number of support points is considered good if the service life deviates less than 1% from the service life from the reference load–notch-strain curve. Table 2 summarises the number of load steps from which this condition is fulfilled.

The linear interpolation connects the results of the load steps with the 1st-degree polynomials. The quality of the service life estimation with linear interpolation depends on the load level. The higher the load is, the more load steps are needed to approximate the load–notch-strain curve with linear interpolation sufficiently well.

The smallest maximum deviations among the examined approaches occur in the regression with the power function. However, the power function converges more slowly than the spline interpolation with the converged solution. Depending on the geometry, load height, and material, it may be that the condition $|N_{var.,n}/N_{var.,200} - 1| \leq 1\%$ is never fulfilled regardless of the number of load steps $n_H$ used. For example, the service life of a flat specimen with $K_t$ = 1.5 and $R_m$ = 400 MPa at a high load under variable amplitude loading converges with the power function to a service life that is 3% less than the service life with 200 directly calculated load steps. This is due to the fact that the regression no longer runs exactly through the FEA results with more than two support points. Thus, the true stresses and strains—i.e., the stresses and strains calculated in the FEA—are not used.

The spline interpolation method converges to the exact solution with the smallest number of load steps $n_H$ for variable amplitude loading. The overview of the approaches on the flat specimen with $K_t$ = 3 and $R_m$ = 800 MPa at a tensile load shows that the spline interpolation is best suited to simulate the load–notch-strain curve.

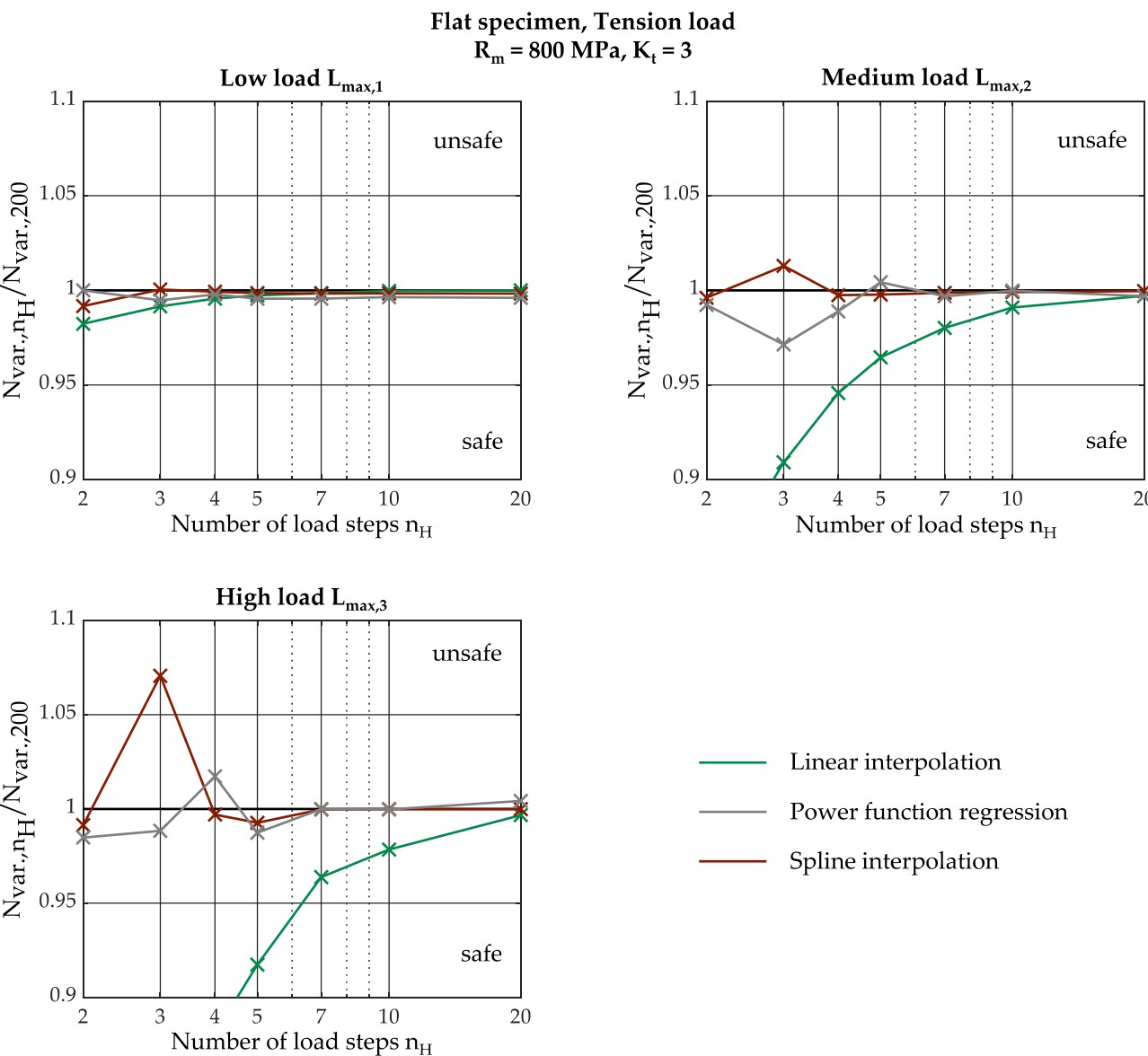

**Figure 19.** Number of load steps $n_H$ for a flat specimen under tension loading with different approaches.

**Table 2.** Number of load steps $n_H$ for a flat specimen under tension loading with different approaches.

| Constraint | Load | Minimum Number of Load Steps $n_H$ | | |
|---|---|---|---|---|
| | | Linear Interpolation | Power Function | Spline Interpolation |
| $\left\|\frac{N_{var.,n_H}}{N_{var.,200}} - 1\right\| \leq 1\%$ | Low $L_{max,1}$ | 3 | 2 | 2 |
| | Medium $L_{max,2}$ | 10 | 5 | 4 |
| | High $L_{max,3}$ | 20 | 7 | 4 |

### 4.6.2. Influence of Material Behaviour and Stress Concentration Factor on the Quality of the Spline Interpolation Method

In Figure 20, the service lives under variable amplitude loading with load–notch-strain curves of a flat specimen with $K_t$ = 1.5, 3, and 5 and $R_m$ = 400, 800, and 1200 MPa under a tensile load are compared with the service lives from reference load–notch-strain curves determined with $n_H$ = 200 load steps. Table 3 again summarises the number of load steps $n_H$ necessary to fulfil criterion $|N_{var.,n}/N_{var.,200} - 1| \leq 1\%$. Figure 20 and Table 3 show that with a higher load, more load steps $n_H$ are necessary. For all cases investigated, a total

of $n_H$ = 5 load steps is sufficient. It can be seen that the deviations increase with lower tensile strengths $R_m$ and lower stress concentration factors $K_t$, and that more load steps $n_H$ are necessary. Accordingly, the largest deviations occur at $R_m$ = 400 MPa and $K_t$ = 1.5.

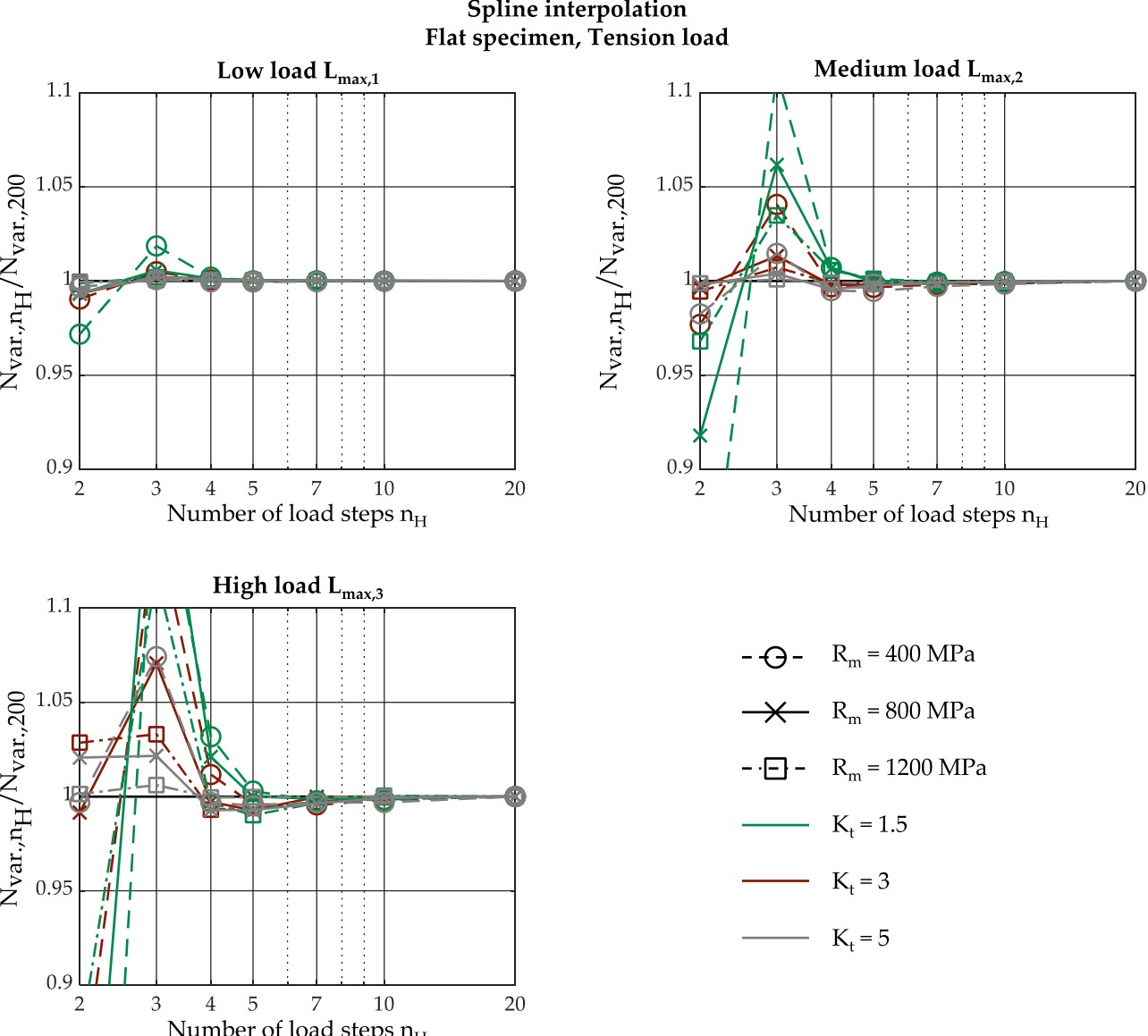

**Figure 20.** Number of load steps $n_H$ for a flat specimen under tension loading with spline interpolation.

**Table 3.** Number of load steps $n_H$ for a flat specimen under tension loading with spline interpolation.

| Constraint | Load | $R_m$ | $K_t$ = 1.5 | $K_t$ = 3 | $K_t$ = 5 |
|---|---|---|---|---|---|
| $\left\lvert \dfrac{N_{var.,n_H}}{N_{var.,200}} - 1 \right\rvert \leq 1\%$ | Low | 400 MPa | 4 | 2 | 2 |
| | | 800 MPa | 2 | 2 | 2 |
| | | 1200 MPa | 2 | 2 | 2 |
| | Medium | 400 MPa | 4 | 4 | 4 |
| | | 800 MPa | 4 | 4 | 2 |
| | | 1200 MPa | 4 | 3 | 2 |
| | High | 400 MPa | 5 | 5 | 4 |
| | | 800 MPa | 5 | 4 | 4 |
| | | 1200 MPa | 5 | 4 | 2 |

### 4.6.3. Influence of Geometry and Load Type on the Quality of the Spline Interpolation Method

In Figure 21 and Table 4, the remaining geometries described in Section 4.2 are examined with the obviously worst case of $R_m = 400$ MPa and $K_t = 1.5$ at a high load. For the planet carrier, the notch sharpness results from the geometry. Additionally, for the other geometries and load types, good service life estimations result from a total of $n_H = 5$ load steps.

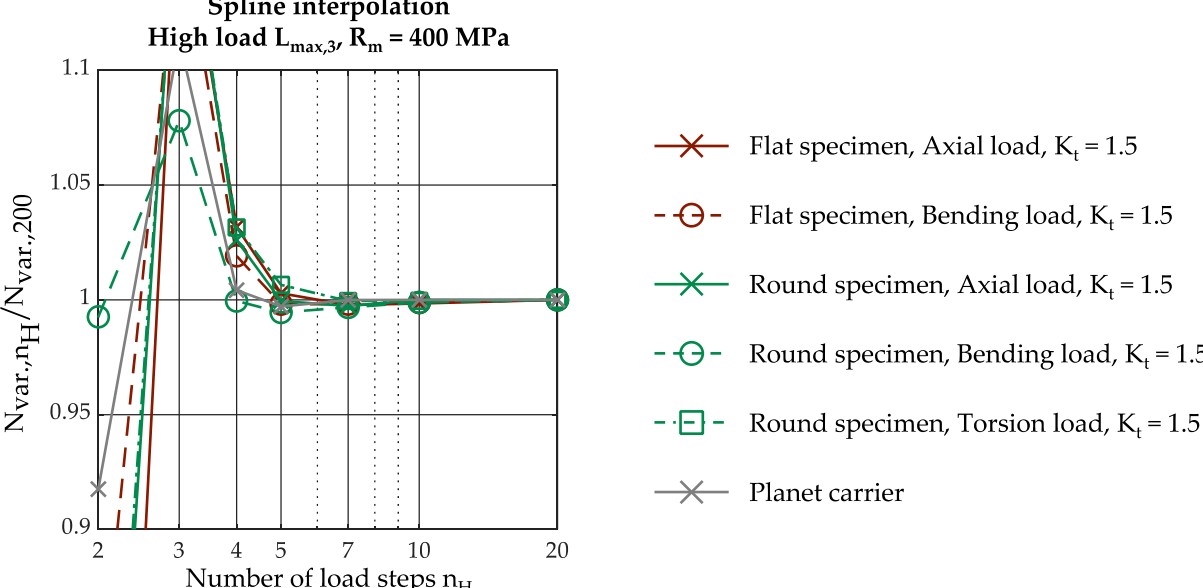

**Figure 21.** Number of load steps $n_H$ for different geometries with spline interpolation.

**Table 4.** Number of load steps $n_H$ for different geometries with spline interpolation.

| Constraint | Specimen | Load | $n_H$ |
|---|---|---|---|
| $\left\| \frac{N_{var,n_H}}{N_{var,200}} - 1 \right\| \leq 1\%$ | Flat specimen | Tension | 5 |
| | | Bending | 5 |
| | Round specimen | Tension | 5 |
| | | Bending | 4 |
| | | Torsion | 5 |
| | Planet carrier | - | 4 |

### 4.7. Summary and Recommendations for the User

In Section 3, methods for improving the efficiency of the determination of load–notch-strain curves in an FEA with elastic-plastic material behaviour were described; these were subsequently examined in Section 4 using a database with regard to their quality and efficiency. In this section, a summary of the results and a resulting recommendation for the user are given.

The correlation between the load–notch-strain curve for the hysteresis branch and the initial load must always be taken into account. With these, the efficiency of the load–notch-strain curve determination can be increased without a loss of quality, as the determination of the load–notch-strain curve for the initial load by means of FEA is no longer necessary.

The material behaviour is entered in the FE software as a discretised curve. The investigations in Section 4.4 show that the number of support points used for the description of the material behaviour influences the quality and efficiency of the load–notch-strain curve determination, and thus also the service life estimation. In addition, a decrease in quality is observed with an increasing load height at the same number of support points. As a good compromise between quality and efficiency, the recommended number of support points for the material law is $n_M = 50$ up to a stress range of twice the tensile strength $\Delta\sigma = 2 \cdot R_m$.

The type of evaluation of the node results in the FEA with elastic-plastic material behaviour has a great influence on the efficiency and quality of the load–notch-strain curve determination. The standard procedure of many FE programs is to copy the results at the Gauss points to the element nodes. An example calculation on a flat specimen notched on both sides shows that the load–notch-strain curves generated in this way do not converge to a constant service life even with fine meshing, and the service lives are on the unsafe side. When using an extrapolation method, as is common in calculations with elastic material behaviour, a converged service life can be achieved in the example even with coarser meshing.

In the case where only the stresses and strains on the surfaces of components are of interest, the use of shell elements on the component surface brings immense advantages for increasing efficiency. Due to the shell elements, Gauss points are available on the component surface, whereby load–notch-strain curves with the same quality can be calculated with a significantly coarser meshing. The simulations on a notched flat specimen show that (at least with the ANSYS Workbench program used) copying the results of the Gauss points to the element nodes is preferable to extrapolation.

In Section 4.6, methods for estimating the load–notch-strain curve from only a few simulated load steps are presented and compared. The linear interpolation method is mentioned only because of its simplicity. In order to exclude the influence of the load–notch-strain curve on the service life, a relatively high number of load steps must be applied in the simulation compared to the other methods, meaning that the efficiency is only increased to a small extent.

The regression of the load–notch-strain curve with the power function has advantages and disadvantages. The advantage is that the load–notch-strain curve is described by a function that is not defined in sections. This means that if the maximum value of the spectrum changes slightly after the FEA is completed, the load–notch-strain curve can also be estimated for other, even slightly higher, maximum values. For sectionwise defined functions, on the other hand, no load–notch-strain curve may be generated for a maximum value of the spectrum that is higher than the maximum applied load in the FEA. The biggest disadvantage of regression with the power function is that the regression does not directly reach the results of the FEA, but only approximates them. As a result, the calculated service life with the estimated load–notch-strain curve does not converge with the service life of the completely simulated load–notch-strain curve.

The best results were achieved with—and, therefore, the recommended method is—the spline interpolation method. In all the cases examined, it is sufficient to apply only five load steps to determine the load–notch-strain curve for the hysteresis branch, as opposed to one load step for each class limit. With the recommended number of classes being 200 for the load–notch-strain curve of the hysteresis branch from the FKM guideline nonlinear, this would mean that 200 simulated load steps are needed.

## 5. Comparative Calculation

Finally, in comparative calculations on a notched specimen, four procedures for determining the service life at variable amplitudes are compared:

1.  As a reference for the calculation time, the load–notch-strain curves are determined as described in Section 2.3 without taking into account the methods described for increasing efficiency in Section 3. The load–notch-strain curves for the initial load and the hysteresis branch are determined separately in FEA. This means that for the 100 classes recommended by the FKM guideline nonlinear [25], $n_I = 100$ load steps are simulated for the load–notch-strain curve of the initial load, and $n_H = 200$ load steps are simulated for the load–notch-strain curve of the hysteresis branch. The material law is modelled with $n_M = 200$ support points. Section 4.5 demonstrated that the procedure of copying the results at the Gauss points to the element nodes does not give a converged result even with a fine mesh. Therefore, stresses and strains at the element nodes are determined by extrapolation, whereby a mesh with

$n_E$ = 30 is sufficient. Outside the notch area, the meshing deviates from Section 4.5; see Figure 15. There, the specimen is meshed with hexahedral elements outside the notch. Tetrahedral elements are used for the comparison, which means that the meshing in the rest of the specimen is almost independent of the meshing in the notch.

2. In Section 2.2, it is claimed that the general use of load–notch-strain curves, as opposed to recalculating the complete load sequence in an FEA with elastic-plastic material behaviour, results in an immense time advantage. To substantiate this, 0.01% of the reversal points of the load sequence from Section 4.1 are applied to the flat specimen and the CP-time is then extrapolated linearly. The section of the load sequence is chosen to include the maximum value of the spectrum; this is shown in Figure 12B. Since the complete load sequence is not calculated, no life estimation can be performed afterwards. The material law is deposited with $n_M$ = 50 interpolation points, and the meshing and evaluation of the node results are selected as in the first case.

3. In the third case, load–notch-strain curves are determined for the service life estimation with the methods mentioned without the use of shell elements on the surface. This means that the load–notch-strain curve for the hysteresis branch is calculated from $n_H$ = 5 load steps with spline interpolation, and the load–notch-strain curve is derived from this. The material law is deposited with $n_M$ = 50 support points, and the meshing and evaluation of the node results are selected as in the first case.

4. In the fourth case, the procedure is the same as that used in the third case, except that shell elements are used on the component surface, which means that a coarser mesh with $n_E$ = 5 can be used.

For these four cases, the load–notch-strain curves and subsequently the service lives are determined on a flat specimen with $R_m$ = 800 MPa and $K_t$ = 3, and a high collective maximum value $L_{max,3}$.

Table 5 summarises the boundary conditions of the individual cases examined. The comparison of the CP-Times is used to evaluate the efficiency. These are related to the CP-Time of the direct load–notch-strain curve determination (1st case). The time taken to perform the spline interpolation is neglected, as this is <0.1 s. It is obvious that due to the calculation time, the calculation reversal point to reversal point cannot be applied for longer load time histories. By using the presented methods without shell elements, the CP-Time can be reduced to 0.023 times the original time. With the use of shell elements, even a reduction to 0.005 times is possible. A comparison of the three calculated service lives (point 1, 3, and 4) shows only a minor deviation. Therefore, the quality of the calculated service lives is independent of the optimisation, but the gain in efficiency is tremendous.

**Table 5.** Summary of comparative calculation.

| Parameter | Direct without Efficiency Methods | Point to Point in FEA | Efficient without Shell Elements | Efficient with Shell Elements |
|---|---|---|---|---|
| Specimen | Flat specimen; $K_t$ = 3 | | | |
| Load | Tension; High load $L_{max,3}$ | | | |
| Tensile strength $R_m$ | 800 MPa | | | |
| Support point material law $n_M$ | 200 | 50 | 50 | 50 |
| Number of Elements per 90° $n_E$ | 30 | 30 | 30 | 5 |
| Method load–notch-strain curve | direct | No load–notch-strain curve | Spline interpolation | Spline interpolation |
| Load steps initial load $n_I$ | 100 | - | - | - |
| Load steps hysteresis branch $n_H$ | 200 | - | 5 | 5 |
| Elements notch root | 27,300 | 27,300 | 27,300 | 120 |
| Elements total | 55,585 | 55,585 | 55,585 | 13,284 |
| CP-Time | 8217 s | $\approx 2 \times 10^8$ s *1 | 188 s | 44 s |
| $\frac{CP-Time}{CP-Time_{direct}}$ | 1 | $\approx$24,000 | 0.023 | 0.005 |
| $\frac{N_{var.}}{N_{var., converged}}$ *2 | 0.985 | - *1 | 0.978 | 1.014 |

*1 Only 283 reversal points (0.01% of the load sequence) are simulated, and the CP-Time is extrapolated linearly. Therefore, no service life can be calculated. *2 The converged service life $N_{var.,converged}$ is taken from the reference calculation provided in Section 4.5.

## 6. Conclusions

The load–notch-strain curve describes the relationship between the load on the component and the locally occurring strain when elastic-plastic material behaviour is taken into account. It is a practical tool for the efficient determination of the local stress–strain path, which is required for a service life estimation according to the notch strain approach. This contribution describes how the load–notch-strain curve can be calculated using an FEA with elastic-plastic material behaviour and how the efficiency of this procedure is increased. The following points must be considered in order to carry out the determination of the load–notch-strain curve for the application of the FKM guideline nonlinear [25] as efficiently and accurately as possible with FEA:

1. The loads in the simulation are applied with a constant distance.
2. It is sufficient to determine the load–notch-strain curve for the hysteresis branch, from which the load–notch-strain curve for the initial load is derived.
3. The material law should be stored in the FE programme with $n_M = 50$ support points.
4. By using shell elements on the surface of the component in the area to be analysed, the Gauss points are shifted to the surface. This makes it possible to use a relatively coarse mesh despite the elastic-plastic material behaviour.
5. When using spline interpolation, it is possible to determine the load–notch-strain curve for the hysteresis branch with only $n_H = 5$ load steps with a good accuracy.

The standard procedure in the FKM guideline nonlinear for determining the load–notch-strain curve is the estimation with the help of a notch root approximation. For both permitted notch root approximations, an FEA with elastic material behaviour must be carried out. In addition, the limit load factor $K_P$ must be determined using an FEA with elastic-ideal-plastic material behaviour. The time required to determine the limit load factor makes the use of notch root approximations less attractive.

In contrast, this contribution shows that the determination of load–notch-strain curves in an FEA with elastic-plastic material behaviour using methods for increasing the efficiency is a good alternative to notch approximation methods in terms of quality and efficiency.

**Author Contributions:** Conceptualisation, L.M., R.B. and M.W.; methodology, L.M.; software, L.M., R.B. and M.W.; validation, L.M.; writing—original draft preparation, L.M.; writing—review and editing, R.B., M.W. and A.E.; visualisation, L.M.; supervision, A.E.; project administration, M.W. and A.E. All authors have read and agreed to the published version of the manuscript.

**Funding:** This research received no external funding.

**Institutional Review Board Statement:** Not applicable.

**Informed Consent Statement:** Not applicable.

**Data Availability Statement:** The data presented in this study are available upon request from the corresponding author.

**Acknowledgments:** We acknowledge financial support by the Open Access Publishing Fund of Clausthal University of Technology.

**Conflicts of Interest:** The authors declare no conflict of interest.

## Appendix A

This section describes the regression used to determine the parameters c and d for the power approach described in Section 3.4.2. As shown in Figure 8, when $\log(\Delta\varepsilon_{FEA} - \Delta\sigma_{el,FEA}/E)$ is plotted against $\log(\Delta\sigma_{el,FEA})$ using a double logarithmic scale, the curve can be described using a linear scale for $\log(\Delta\varepsilon_{FEA} - \Delta\sigma_{el,FEA}/E) \geq -4$. For the linear regression, the logarithm of Equation (A1) and the support points from the FEA are used; see Equation (A2). The substitutions in Equations (A3)–(A6) result in a linear equation in the form of Equation (A7).

$$\Delta\varepsilon_P(\Delta\sigma_{el}) = \frac{\Delta\sigma_{el}}{E} + 2\left(\frac{\Delta\sigma_{el}}{2\cdot c}\right)^{\frac{1}{d}} \qquad\qquad (A1)$$

$$\log\left(\Delta\varepsilon_{\text{FEA}} - \frac{\Delta\sigma_{\text{el,FEA}}}{E}\right) = \frac{1}{d} \cdot \log(\Delta\sigma_{\text{el,FEA}}) - \frac{1}{d} \cdot \log(2 \cdot c) + \log(2) \tag{A2}$$

$$y = \log\left(\Delta\varepsilon_{\text{FEA}} - \frac{\Delta\sigma_{\text{el,FEA}}}{E}\right) \tag{A3}$$

$$a = \frac{1}{d} \tag{A4}$$

$$x = \log(\Delta\sigma_{\text{el,FEA}}) \tag{A5}$$

$$b = -\frac{1}{d} \cdot \log(2 \cdot c) + \log(2) \tag{A6}$$

$$y = a \cdot x + b \tag{A7}$$

The parameters a and b are obtained by the method of least squares using Equations (A8) and (A9) and n pairs of values x and y:

$$a = \frac{n \cdot \sum_{i=1}^{n} x_i \cdot y_i - \sum_{i=1}^{n} x_i \cdot \sum_{i=1}^{n} y_i}{n \cdot \sum_{i=1}^{n} x_i^2 - \left(\sum_{i=1}^{n} x_i\right)^2} \tag{A8}$$

$$b = \frac{\sum_{i=1}^{n} x_i^2 \cdot \sum_{i=1}^{n} y_i - \sum_{i=1}^{n} x_i \cdot \sum_{i=1}^{n} x_i \cdot y_i}{n \cdot \sum_{i=1}^{n} x_i^2 - \left(\sum_{i=1}^{n} x_i\right)^2} \tag{A9}$$

The parameters c and d are obtained by transforming Equations (A4) and (A6):

$$d = \frac{1}{a} \tag{A10}$$

$$c = \frac{1}{2} \cdot 10^{\frac{1}{a} \cdot (\log(2) - b)} \tag{A11}$$

In the regression, only pairs of values from $\Delta\sigma_{\text{el,FEA}}$ and $\Delta\varepsilon_{\text{FEA}}$ from FEA that fulfil the condition $\log(\Delta\varepsilon_{\text{FEA}} - \Delta\sigma_{\text{el,FEA}}/E) \geq -4$ are considered. In the case that less than two load steps are necessary to perform a linear regression to fulfil the condition $\log(\Delta\varepsilon_{\text{FEA}} - \Delta\sigma_{\text{el,FEA}}/E) \geq -4$, all load steps for which $\Delta\varepsilon_{\text{FEA}} - \Delta\sigma_{\text{el,FEA}}/E > 0$ are used. This last restriction is necessary because in the range of low loads, a negative difference $\Delta\varepsilon_{\text{FEA}} - \Delta\sigma_{\text{el,FEA}}/E$ can occur due to numerical inaccuracies during the FEA. Since the logarithm can only be applied to values greater than zero, only data pairs for which $\Delta\varepsilon_{\text{FEA}} - \Delta\sigma_{\text{el,FEA}}/E > 0$ applies are considered.

**Appendix B**

In this section, the determination of the parameters $a_i$, $b_i$, $c_i$, and $d_i$ for the spline interpolation in Section 3.4.3 is described. Boundary conditions are defined for the individual sections of the function: to avoid jumps occurring in the load–notch-strain curve, the strain ranges at the beginning of each section must match those at the end of the previous section, and the strain ranges at the end of each section must match those at the beginning of the following section; see Equations (A12) and (A13). These conditions correspond to the boundary conditions of interpolation with a polygonal chain; see Section 3.4.1. In addition, Equation (A14) guarantees a kink-free transition at the support points, and Equation (A15) generates a two-times continuously differentiable function:

$$\Delta\varepsilon_{S,i}(\Delta\sigma_{\text{el,FEA},i}) = \Delta\varepsilon_{\text{FEA},i} \qquad \text{with } i = 1, \ldots, n_H \tag{A12}$$

$$\Delta\varepsilon_{S,i}(\Delta\sigma_{\text{el,FEA},i+1}) = \Delta\varepsilon_{\text{FEA},i+1} \qquad \text{with } i = 1, \ldots, n_H \tag{A13}$$

$$\Delta\varepsilon'_{S,i}(\Delta\sigma_{\text{el,FEA},i+1}) = \Delta\varepsilon'_{S,i+1}(\Delta\sigma_{\text{el,FEA},i+1}) \qquad \text{with } i = 1, \ldots, n_H \tag{A14}$$

$$\Delta\varepsilon''_{S,i}(\Delta\sigma_{\text{el,FEA},i+1}) = \Delta\varepsilon''_{S,i+1}(\Delta\sigma_{\text{el,FEA},i+1}) \qquad \text{with } i = 1, \ldots, n_H \tag{A15}$$

In order to determine the parameters $a_i$, $b_i$, $c_i$, and $d_i$ with a linear system of equations, two further boundary conditions must be defined. At the left edge of the load–notch-strain curve, the modulus of elasticity E is specified as the gradient; at the right edge, the load–notch-strain curve should end with a point of inflection. This results in:

$$\Delta\varepsilon'_{S,1}(\Delta\sigma_{el,FEA,1}=0) = \frac{1}{E} \tag{A16}$$

$$\Delta\varepsilon''_{S,n}(\Delta\sigma_{el,FEA,n_H+1}) = 0 \tag{A17}$$

With the linear system of equations in Equation (A18), the vector **b** containing the parameters $b_i$ is calculated:

$$\mathbf{A} \cdot \mathbf{b} = \mathbf{g} \tag{A18}$$

The matrix **A** and the vectors **b** and **g** are defined as follows:

$$h_i = \Delta\sigma_{el,FEA,i+1} - \Delta\sigma_{el,FEA,i} \qquad \text{with } i = 1,\ldots,n_H \tag{A19}$$

$$\mathbf{A} = \begin{bmatrix} 1 & 0.5 & & & \\ h_1 & 2(h_1+h_2) & h_2 & & \\ & & \ddots & & \\ & & h_{n_H-1} & 2(h_{n_H-1}+h_{n_H}) & h_{n_H} \\ & & & & 1 \end{bmatrix} \tag{A20}$$

$$\mathbf{b} = \begin{bmatrix} b_1 \\ b_2 \\ \vdots \\ b_{n_H} \\ 0 \end{bmatrix} \tag{A21}$$

$$\mathbf{g} = \begin{bmatrix} g_1 \\ g_2 \\ \vdots \\ g_{n_H} \\ 0 \end{bmatrix} \tag{A22}$$

$$g_1 = \frac{3}{2}\left(\frac{\Delta\varepsilon_2 - \Delta\varepsilon_1}{h_1{}^2} - \frac{1}{E \cdot h_1}\right) \tag{A23}$$

$$g_i = 3\left(\frac{\Delta\varepsilon_{i+1} - \Delta\varepsilon_i}{h_i} - \frac{\Delta\varepsilon_i - \Delta\varepsilon_{i-1}}{h_{i-1}}\right) \quad \text{with } i = 2,\ldots,n_H \tag{A24}$$

The remaining parameters $a_i$, $c_i$, and $d_i$ are calculated with Equations (A25)–(A27):

$$a_i = \frac{b_{i+1} - b_i}{3\,h_i} \qquad \text{with } i = 1,\ldots,n_H \tag{A25}$$

$$c_i = \frac{b_{i+1} - b_i}{h_i} - \frac{2\,b_i + b_{i+1}}{3}\,h_i \qquad \text{with } i = 1,\ldots,n_H \tag{A26}$$

$$d_i = \Delta\varepsilon_i \qquad \text{with } i = 1,\ldots,n_H \tag{A27}$$

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
