# Peer review of "Determination of Local Stresses and Strains within the Notch Strain Approach: The Efficient and Accurate Calculation of Notch Root Strains Using Finite Element Analysis"

_applsci, doi:10.3390/app112411656_

Round 1

Reviewer 1 Report

Firstly, the reviewer would like to appreciate the authors for the comprehensive contents covered and the clear structure of the manuscript. The authors present a procedure for determining load–notch-strain curves using FEA efficiently, a comparative calculation is also carried out. The reviewer thinks the work is useful, comprehensive and interesting, and suggests the manuscript can be received with a minor revision.

Some minor comments are listed below:

- It will be helpful if the authors can translate ref [25,26] into English in the future to help the researchers all over world to get familiar the concept and the procedure.

- What is the significant difference between this manuscript and ref [33]?

- The author may want to comments on how to deal with the “sharp notch”, namely, geometry discontinuity, at which location the stress/strain may be singular.

- The traditional Ramberg Osgood equation (3) is based on total strain, and the second term in Eq.(3) suggests that the nonlinear behavior has already developed even though the stress is very small (far beyond the yield stress), can the authors make some comments on this behavior?

- The literature review is well written, but it will be interesting if the authors can discussed some other stress and strain based concept for assessment fatigue behavior of structures. These work also deal with FEM determination of strain parameters considering the material elastic-plastic behavior, some typical literatures like “Fatigue behavior of typical details of orthotropic steel bridges in multiaxial stress states using traction structural stress”,  “Fatigue Performance of Different Rib-To-Deck Connections Using Traction Structural Stress Method”, “A structural strain parameter for a unified treatment of fatigue behaviors of welded components”, “An analytically formulated structural strain method for fatigue evaluation of welded components incorporating nonlinear hardening effects”

Reviewer 2 Report

The article systematically discussed the effects of related factors on the load-notch-strain curve determination in the framework of FKM guideline nonlinear, and optimization suggestions are proposed. The context is well organized and fully discussed. It is recommended to be published. However, some minor comments and questions are listed below for your reference.

1. [Line 676] Is the nM discussed in Section 4.4 related to the structure configurations or the mesh quality of the model? It is necessary to add more discussions similar to Section 4.6.

2. [Line 706] Why the load-notch-strain curve determined using shell elements on the surface and the number of elements per 90° nE=70 is chosen as the reference? How about the predicted life based on the reference load-notch-strain curve compared with the experiment results? Is it appropriate to use the predicted life based on the reference load-notch-strain curve to discuss the conservatism of the other optimizations?

3. Is this method suitable for the life prediction of components subject to nonproportional multiaxial cyclic loadings? Explainations should be provided for this point.

Reviewer 3 Report

This is a good approach to problem derived from notch and grooves in some critical parts, the gearbox planet carrier is a good example, and a high solicited part in modern windmills.

I know the part well, and it is described in Machines 2017, 5(2), 15; https://doi.org/10.3390/machines5020015 where a complete study of 10 MW case was presented, including the very difficult finishing of the part. You can include in the state of the art. Some references are very old, from 1977

Appendixes are very good.

In short, make a wider view of planet carrier in gearboxes 8above is one key) and paper could be OK.
